# Deciphering caveolar functions by *syndapin III* KO-mediated impairment of caveolar invagination

Eric Seemann[1†], Minxuan Sun[1†], Sarah Krueger[1], Jessica Tröger[1], Wenya Hou[1], Natja Haag[1], Susann Schüler[1], Martin Westermann[2], Christian A Huebner[3], Bernd Romeike[4], Michael M Kessels[1]*, Britta Qualmann[1]*

[1]Institute for Biochemistry I, Jena University Hospital – Friedrich Schiller University Jena, Jena, Germany; [2]Electron Microscopy Center, Jena University Hospital – Friedrich Schiller University Jena, Jena, Germany; [3]Institute for Human Genetics, Jena University Hospital – Friedrich Schiller University Jena, Jena, Germany; [4]Institute of Pathology, Division of Neuropathology, Jena University Hospital – Friedrich Schiller University Jena, Jena, Germany

**Abstract** Several human diseases are associated with a lack of caveolae. Yet, the functions of caveolae and the molecular mechanisms critical for shaping them still are debated. We show that muscle cells of *syndapin III* KO mice show severe reductions of caveolae reminiscent of human caveolinopathies. Yet, different from other mouse models, the levels of the plasma membrane-associated caveolar coat proteins caveolin3 and cavin1 were both not reduced upon *syndapin III* KO. This allowed for dissecting bona fide caveolar functions from those supported by mere caveolin presence and also demonstrated that neither caveolin3 nor caveolin3 and cavin1 are sufficient to form caveolae. The membrane-shaping protein syndapin III is crucial for caveolar invagination and KO rendered the cells sensitive to membrane tensions. Consistent with this physiological role of caveolae in counterpoising membrane tensions, syndapin III KO skeletal muscles showed pathological parameters upon physical exercise that are also found in *CAVEOLIN3* mutation-associated muscle diseases.
DOI: https://doi.org/10.7554/eLife.29854.001

*For correspondence:
Michael.Kessels@med.uni-jena.de (MMK);
Britta.Qualmann@med.uni-jena.de (BQ)

†These authors contributed equally to this work

**Competing interests:** The authors declare that no competing interests exist.

## Introduction

Caveolae - uniform plasma membrane invaginations with ~70 nm diameter - were first described more than 60 years ago (*Yamada, 1955*). Yet, their functions are still matter of debate. For decades, they were considered as membrane trafficking compartments. More recent data suggested that caveolae may rather represent signaling platforms and/or mechanosensors (*Nassoy and Lamaze, 2012*; *Parton and del Pozo, 2013*; *Shvets et al., 2014*; *Cheng and Nichols, 2016*).

A main structural component of caveolar coats in muscles is caveolin3 (cav3). *CAV3* mutations manifest in several human diseases, for example limb girdle muscular dystrophy (LGMD), rippling muscle disease (RMD), hyperCK(creatine kinase)emia, distal myopathy, hypertrophic cardiomyopathy, arrhythmogenic-long-QT syndrome and sudden-infant-death syndrome. Most patients with *CAV3* mutations are heterozygous and the pathophysiology seems to be caused by the mutated protein acting dominant-negatively on WT-cav3 via self-association (*Gazzerro et al., 2010*). Consistently, also overexpression of dominant-negative mutants in mice caused symptoms resembling hypertrophic cardiomyopathy (*Ohsawa et al., 2004*).

*Cav3* KO led to a lack of caveolae, size variability of muscle fibers and cases of necrosis considered as signs of muscle dystrophy (*Hagiwara et al., 2000*; *Galbiati et al., 2001*). *Cav3* KO hearts

were unaltered in one study (*Galbiati et al., 2001*), whereas another reported heart hypertrophy and dilation (*Woodman et al., 2002*). In zebrafish, cav3 knock-down caused notochord and myoblast fusion defects and impaired movement (*Nixon et al., 2005*). Cav3 interacts with a variety of signaling components and may also have roles in energy metabolism. It is therefore unclear whether the broad effects of *CAV3* deficiency reflected by the different diseases and the in part contradictory effects of *cav3* mutants are caused by aberrant signaling or caveolar dysfunctions. Similar limitations hampered the interpretations of cav1 analyses (*Gazzerro et al., 2010*; *Le Lay and Kurzchalia, 2005*).

Additionally, a variety of examinations of human mutations argue against simple correlations of caveolin availability and disease phenotypes or against simply correlating caveolae numbers and clinical symptoms. Patients with heterozygous V57M exchange showed hyperCKemia, yet, cav3 levels only were reduced by 62% (*Alias et al., 2004*). The heterozygous disease mutations CAV3 P28L and R27E did not lead to any significant reduction in caveolae density in quantitative electron microscopical analyses of patient biopsies (*Timmel et al., 2015*). Even in LGMD-1C patients with the severest cases of *CAV3* missense mutations known (63TFT65del and P104L) showing full clinical symptoms, caveolae formation was not fully abolished but 7 or 24% of WT remained, as judged from the representative images presented (*Minetti et al., 1998*; *Minetti et al., 2002*).

Furthermore, the observation that the same *CAV3* mutation can lead to different clinical phenotypes suggests additional, maybe indirect mechanisms (*Fischer et al., 2003*). Extensive quantitative analyses demonstrated that major portions of both of the endogenous non-muscle caveolins, cav1 and cav2, were not located at deeply invaginated classical caveolar profiles (*Fujimoto et al., 2000*). It thus seems possible that some *CAV* phenotypes may be linked to caveolins acting as membrane-associated scaffolds rather than to caveolae as such. Apart from such putatively distinct functions of caveolin, cav1-caveolar invaginations were highlighted as membrane tension buffers in a recent seminal paper (*Sinha et al., 2011*). Further, most recent papers support this view (*Lo et al., 2015*; *Cheng et al., 2015*).

Thus far, however, it has been impossible to clearly distinguish caveolin and caveolar functions. Caveolin-deficiencies obviously are unable to distinguish between these possibilities, as they lack both invaginated caveolae and caveolins (*Hansen and Nichols, 2010*; *Nassoy and Lamaze, 2012*; *Parton and del Pozo, 2013*; *Shvets et al., 2014*; *Cheng and Nichols, 2016*). Unfortunately, the same is true for deficiency of CAVIN-1, a caveolae-associated protein, which forms a flexible, net-like protein mesh around caveolin complexes (*Stoeber et al., 2016*) and has been introduced as factor critical for caveolae formation and function. *CAVIN-1* KO phenotypically mimicked caveolin deficiency merely due to a concomitant, massive decrease of caveolin protein levels (*Hill et al., 2008*; *Liu and Pilch, 2008*; *Liu et al., 2008*).

We here describe that KO of *syndapin III* (also called *PACSIN3*), which encodes for the muscle-enriched isoform of the syndapin family of F-BAR proteins (*Kessels and Qualmann, 2004*; *Qualmann et al., 2011*), impairs caveolar invagination without affecting cav3 plasma membrane levels. This represents the desired possibility to dissect the physiological importance of caveolae from that of cav3 and to thereby reveal the cell biological and physiological functions of caveolar invaginations by comparing the *syndapin III* KO phenotypes to those attributed to cav3 deficiency. Our analyses unveil that - whereas some, mostly cardiac caveolinopathy phenotypes seem unrelated to impairments of caveolar invagination – in particular cellular integrity under strong mechanical stress was significantly affected in *syndapin III* KO muscles. Under physical exercise, failure to invaginate caveolae using the membrane-shaping protein syndapin III coincided with a widened caliber spectrum, detached nuclei and signs of inflammation and necrosis. This pathophysiology in *syndapin III* KO muscles is reminiscent of human myopathies associated with *CAV3* mutation. Thus, syndapin III is crucial for caveolar invagination and thereby highlights that the physiological function of cav3-coated caveolae is to preserve muscle cell integrity upon acute membrane tensions, as they occur during physical exercise.

## Results

### Generation of syndapin III KO mice

Caveolins are required for caveolae formation. Bacterial reconstitutions, furthermore, suggested that caveolins also were sufficient for caveolar invagination (*Walser et al., 2012*). In this case, discrimination between caveolin and bona fide caveolar functions would be impossible. Recent observations, however, suggested that it might nevertheless be possible to unveil specifically the functions of caveolar invaginations. RNAi against the F-BAR protein syndapin II led to a reduction of deeply invaginated caveolae without affecting the presence of endogenous cav1 at the plasma membrane of NIH3T3 cells (*Koch et al., 2012*) and to increased levels of overexpressed cav1 in the TIRF-zone of HeLa cells (*Hansen et al., 2011*), respectively.

Toward addressing the role of caveolar invaginations in the various diseases associated with *CAV3* mutations, we therefore studied the muscle-enriched member of the syndapin family (*Kessels and Qualmann, 2004*; *Qualmann et al., 2011*), syndapin III (*Figure 1a*), by generating *syndapin III* KO mice. The Cre/lox system was used to delete exons 5 and 6 leading to a premature stop. A mouse line lacking *syndapin III* was obtained via mating with constitutively and ubiquitously Cre recombinase-expressing mice (*Figure 1b*).

As schematically depicted in *Figure 1a*, any putative *syndapin III* translation product in the *syndapin III* KO line would be limited to a peptide comprising only the first 70 amino acids of syndapin III and five non-related amino acids. The anti-syndapin III isoform-specific antibodies raised against the non-F-BAR part of the protein (*Koch et al., 2011*) are not able to recognize such a putative peptide. Additional control examinations conducted clearly showed that the putative syndapin III$^{1-70}$ peptide anyhow did not show any F-BAR domain functionalities, such as induction of membrane tubules (*Figure 1c–e*; *Figure 1—figure supplement 1*) and membrane binding (*Figure 1f–h*). Also, the putative syndapin III$^{1-70}$ peptide did not interfere with key syndapin III F-BAR functions, as demonstrated by coexpression. GFP-syndapin III F-BAR still showed its normal distribution (*Figure 1i*; *Figure 1—figure supplement 2*). The observed lack of functionality is in line with the fact that such a putative peptide would only represent a minor fragment of the N-terminal F-BAR domain of syndapin III. *Syndapin III* exons 5,6 deletion thus leads to full syndapin III loss-of-function (*Figure 1a–k*).

*Syndapin III* KO mice were viable, developed without any obvious impairments and were fertile. Litter of heterozygous *syndapin III* KO mice showed normal Mendelian and gender distributions (*Figure 1l*). Analyses at both the mRNA and the protein level confirmed the successful KO of *syndapin III* (*Figure 1m–o*).

Furthermore, immunoblotting of tissue homogenates of WT and *syndapin III* KO mice firmly proved the specificity of the anti-syndapin III immunodetection. Anti-syndapin III immunoblotting exclusively detected syndapin III, as demonstrated by the lack of signal in *syndapin III* KO tissues (*Figure 1o*). Despite the fact that syndapin I and II also are expressed in some of the tissues analyzed (*Kessels and Qualmann, 2004*; *Modregger et al., 2000*), no further bands besides syndapin III were detected by the anti-syndapin III antibodies (*Figure 1o*).

Specific anti-syndapin III immunolabeling was also detected in immunofluorescence analyses of muscle tissues. *Syndapin III* KO tissues showed no anti-syndapin III immunodetections. Analyses of WT samples showed that syndapin III is enriched at the cav3-positive plasma membranes in sections of skeletal muscles (*Figure 1p–r*). Colocalization in transversal tissue sections hereby was very high at the plasma membrane (Pearson correlation coefficient 0.55 ± 0.01) and not existing at all in intracellular areas (Pearson correlation coefficient 0.08 ± 0.01), respectively (*Figure 1q*).

### Syndapin III is crucial for invagination of cav3-coated caveolae

We next subjected primary cardiomyocytes from hearts of WT and *syndapin III* KO mice to fixation, thin sectioning and transmission electron microscopy (TEM). Cardiomyocytes from newborn WT pups showed plasma membrane areas with high frequencies of profiles resembling those of caveolae (*Figure 2a,a'*). In contrast, the plasma membranes of cardiomyocytes from *syndapin III* KO mice largely lacked deeply invaginated structures (*Figure 2b,b'*).

Quantitative analyses of plasma membrane stretches from these sections suggested that indeed deeply invaginated membrane profiles with caveolar appearance were largely absent from *syndapin*

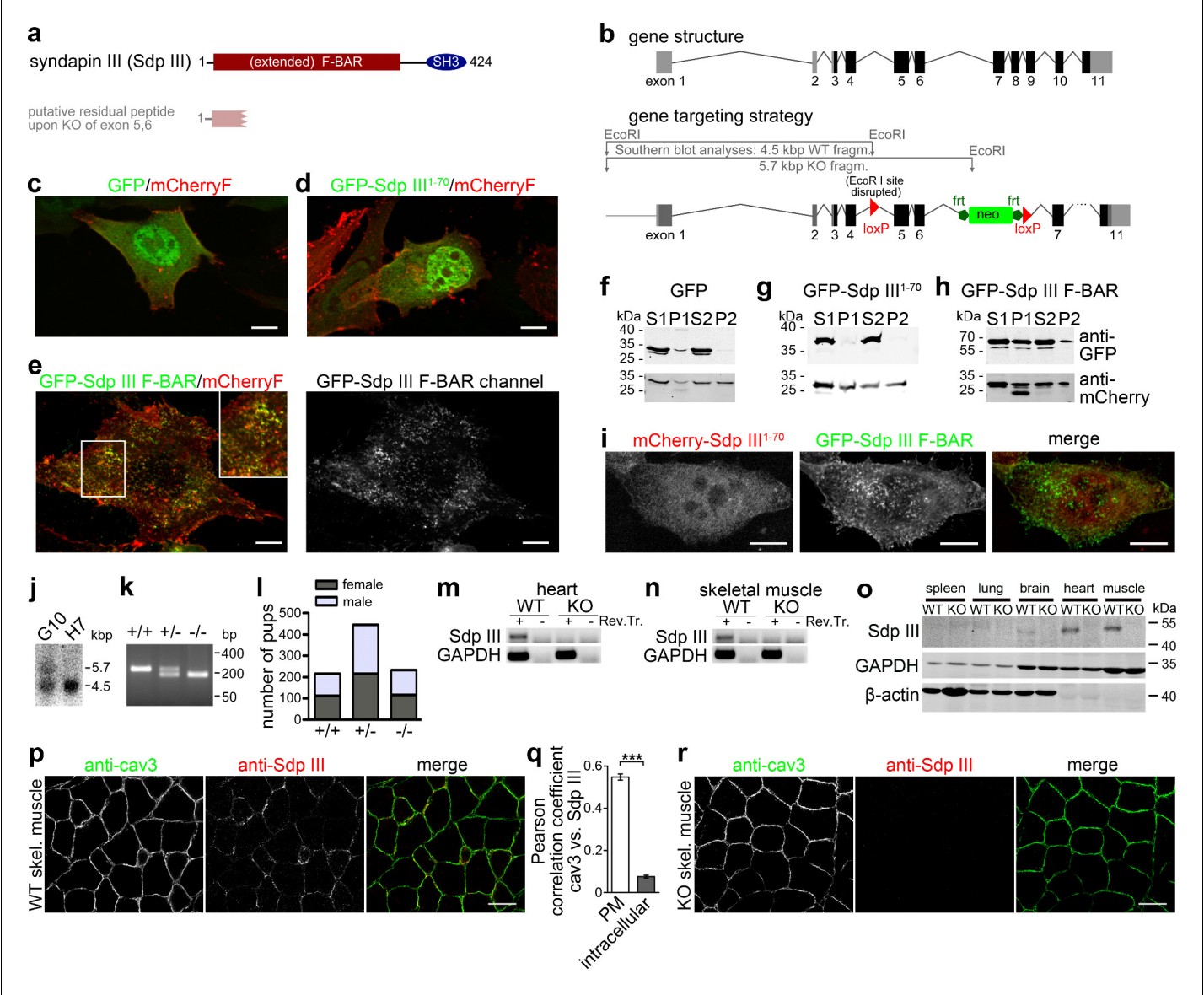

**Figure 1.** Generation of *syndapin III* KO mice. (a) Murine syndapin III domain structure and putative remaining peptide upon *syndapin III* exon 5 and 6 deletion. (b) Generation of *syndapin III* KO mice. Scheme of the *syndapin III* gene comprising 11 exons (coding exon parts in black) and of targeting vector and strategy of Southern blot analyses. Homologous recombination (homologous regions in dark grey and black) resulted in floxed exons 5 and 6. (c–e) Merges of MIPs of NIH3T3 cells transfected with GFP (c), GFP-syndapin III$^{1-70}$ peptide (putatively remaining upon KO; composed of aa1-70 of syndapin III and five unrelated aa resulting from the frameshift caused by exon 5,6 deletion) (d) and GFP-syndapin III F-BAR (e), respectively. Cotransfected plasma membrane-targeted mCherry (mCherryF) served as internal control for a membrane-bound protein. In (e), the GFP channel of a GFP-syndapin III F-BAR-transfected cell is shown in addition to the merge to visualize the tubular structures induced by syndapin III F-BAR. The inset in (e) shows an enlargement of the boxed area. Bars, 10 µm. For enlarged images see *Figure 1—figure supplement 1*. (f–h) Immunoblotting analyses of fractionations of transfected HEK293 cells showing that whereas plasma membrane-targeted mCherry (f–h, lower panel) and GFP-syndapin III F-BAR (h, upper panel) are readily detectable in the crude membrane fraction P2, both GFP and GFP-syndapin III$^{1-70}$ are not (f,g, upper panels). (i) MIPs of NIH3T3 cells transfected with GFP-syndapin III F-BAR coexpressing mCherry-syndapin III$^{1-70}$ showing undisturbed membrane localization, self-assembly and membrane tubulation abilities of GFP-syndapin III F-BAR. Bars, 10 µm. For enlarged images see *Figure 1—figure supplement 2*. (j) Southern blot analysis of exemplary ES cell clones (G10, transgenic; H7, WT). (k) Genotyping of the offspring of heterozygous mating identifies all possible genotypes. (l) Normal frequency of genotypes and genders of *syndapin III* KO mice. (m,n) RT-PCRs on heart (m) and skeletal muscle cDNA (n). (o) Immunoblottings of tissue homogenates (50 µg each) show the lack of syndapin III in KO tissues. GAPDH and β-actin, controls. (p–r) Immunofluorescence analyses of syndapin III and cav3 in transversal skeletal muscle sections from WT (p) and *syndapin III* KO mice (r) and quantitative colocalization analyses (q) in ROIs placed at plasma membrane (n = 240 ROIs from eight images) and intracellular areas (n = 160 ROIs from eight images) in confocal stacks of images of transversal sections of WT skeletal muscles. Data, mean ± SEM. Statistical significance, Mann-Whitney U test. Bars, 100 µm.

*Figure 1 continued on next page*

*Figure 1 continued*

DOI: https://doi.org/10.7554/eLife.29854.002

The following source data and figure supplements are available for figure 1:

**Source data 1.** This spreadsheet contains the data for all quantitative evaluations shown in the different panels of *Figure 1* (i.e. of *Figure 1l and q*).
DOI: https://doi.org/10.7554/eLife.29854.005

**Figure supplement 1.** The syndapin III F-BAR domain localizes to the plasma membrane and decorates distinct membrane domains (enlarged images *Figure 1c–d*).
DOI: https://doi.org/10.7554/eLife.29854.003

**Figure supplement 2.** A putative syndapin III[1-70] peptide does not interfere with syndapin III F-BAR domain functions (enlargement of images shown in *Figure 1i*).
DOI: https://doi.org/10.7554/eLife.29854.004

*III* KO cardiomyocytes. WT membranes showed about 2 caveolae/μm membrane stretch. *Syndapin III* KO mice displayed 0.29 caveolae/μm, that is only 15% of WT (*Figure 2c*).

Immunoblottings with anti-syndapin I antibodies demonstrated that the crucial role of syndapin III in membrane invagination was not compensated for by ectopic expression of the mostly neuronal syndapin I (*Qualmann et al., 1999*) in heart or skeletal muscles (*Figure 2d*). Quantitative western blotting analyses of syndapin II showed that also the expression levels of the more ubiquitously expressed syndapin family member, syndapin II (*Qualmann and Kelly, 2000*), remained unchanged in both heart and skeletal muscles (*Figure 2e–h*). Thus, the identified *syndapin III* KO phenotype specifically and exclusively reflected a loss of the functions of syndapin III and not of other members of the syndapin family.

Thin sectioning of fixed cells is a standard technique, but we were concerned that several caveats of this classical method may compromise our data. First, lipids cannot be fixed well chemically and caveolae are known to be induced, morphologically changed and/or promoted by chemical fixation (*Severs, 1988*) shedding doubt on the numbers and caveolar morphologies we obtained using chemically fixed samples. Second, the membrane profiles we observed were not always clearly identifiable as caveolae. Therefore, especially in WT samples, many of them had to be excluded from the quantitation (*Figure 2a,a'*; question marks). Unambiguous identification of caveolar structures would require 3D-information and immunolabeling. However, even serial sectioning using 50 nm thin sections and 3D-reconstruction would fail to provide reliable 3D-information on 70-nm wide structures. Furthermore, immunolabeling of thin sections is inefficient, as only the section surfaces are accessible for immunodetection. Indeed, even in cases of exceptionally high labeling efficiencies reported, anti-caveolin immunolabeling of sections with caveolae selected for superimposition analyses only reached an average of 2.5 gold particles/caveola (*Ludwig et al., 2013*). Unlabeled caveolae would severely compromise our quantitative analyses. Third, even if caveolae are efficiently recognized, quantitative analyses of membrane topologies would be restricted to areas defined by the length of a membrane segment multiplied by the thickness of the ultrathin section (50 nm), that is would merely represent 0.05 μm² area per μm membrane stretch evaluated (*Figure 2i*, colored lines). Furthermore, we noticed WT plasma membrane stretches without any caveolar profiles, whereas others showed accumulations (*Figure 2a,a'*). Caveolar frequency thus also largely depends on the random orientation of sections in relation to caveolae-enriched regions (*Figure 2i*).

In order to i) preserve membrane topologies by rapid freezing, ii) image perpendicular views of large plasma membrane fields, iii) obtain 3D-information and iv) be able to probe wide areas of the plasma membrane with antibodies, we therefore established immunogold labeling protocols for freeze-fractured primary cardiomyocytes. The P-halves of the fractured membranes provide full antibody access to the entire cytosolic membrane surface (P-face). Labeling densities can thus reach high values. This will ensure the identification of most caveolar membrane profiles as caveolae. The cooling rates used exceeded 4000 K/s, that is membrane topologies were preserved within milliseconds, whereas chemical fixation takes minutes, that is >100,000 times longer. TEM analyses of such membranes provided perpendicular views onto large membrane fields and rotation shadowing provided 3D-information (*Figure 3a,b*).

P-faces of freeze-fractured membrane replica from WT and *syndapin III* KO mice showed specific anti-syndapin III immunogold labeling in WT samples. Specificity controls included incubations of

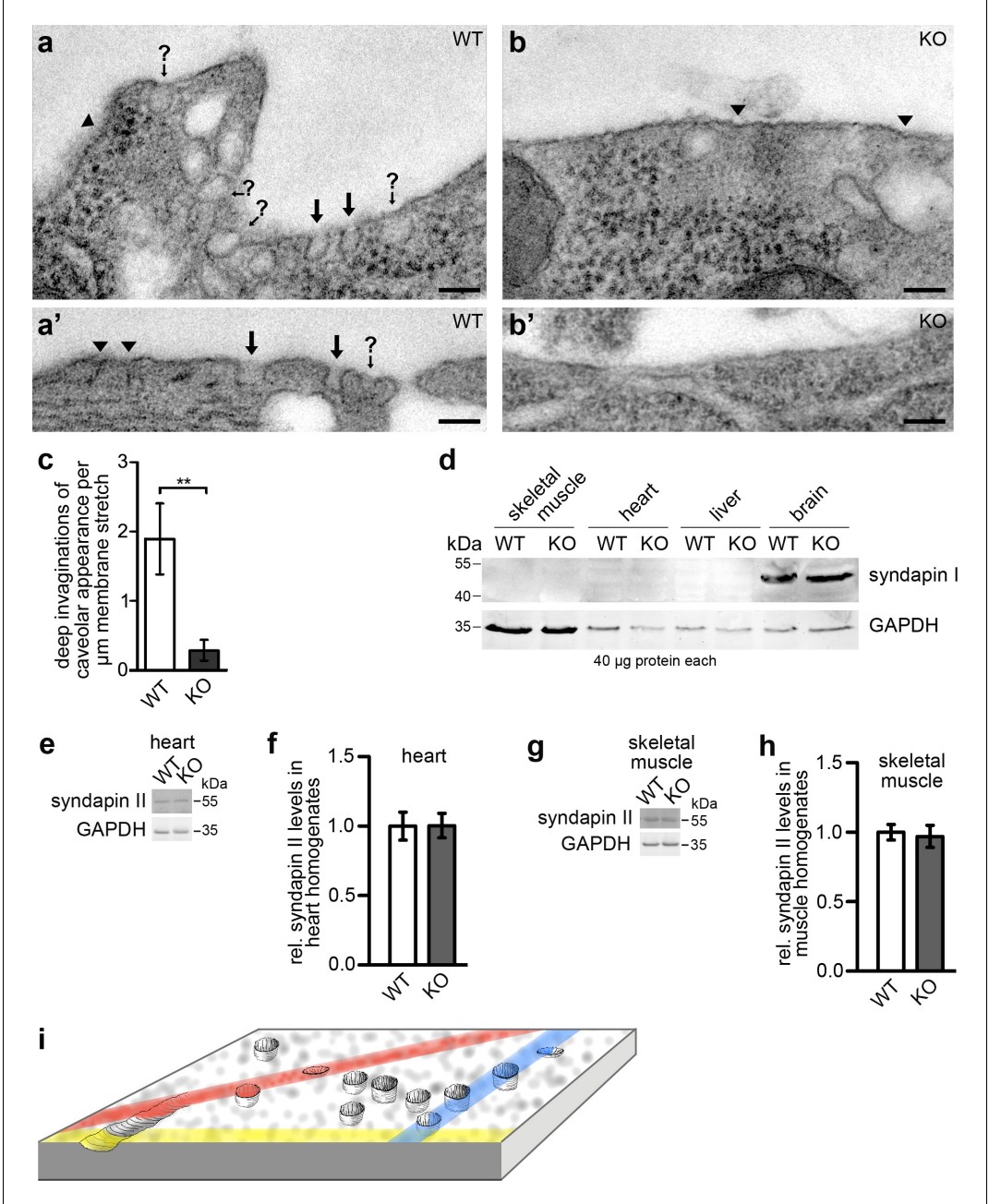

**Figure 2.** *Syndapin III* KO leads to a loss of plasma membrane invaginations with caveolar morphology. (**a–b'**) TEM of 50 nm sections of chemically fixed primary cardiomyocytes isolated from WT (**a,a'**) and *syndapin III* KO mice (**b,b'**), respectively. Marked are deep membrane invaginations hit by the orientation of the section in a way that they can be recognized as caveolar profiles (black arrows), deep invaginations with (often due to non-perpendicular sectioning) unclear opening (arrows with question marks) and more shallow membrane indentations of unclear nature (arrowheads). Note that *syndapin III* KO membrane stretches (**b,b'**) have fewer invaginations. Bars, 100 nm. (**c**) Quantitative analyses of plasma membrane stretches for the presence and frequency of deeply invaginated profiles with caveolar appearance corresponding to structures in images marked with arrows. Data, mean ± SEM. Statistical significance, two-tailed Student's t test. WT, eight membrane stretches; KO, 13 membrane stretches from different cells. (**d**) Immunoblotting analyses of tissue homogenates from WT and *syndapin III* KO mice with anti-syndapin I antibodies show expression of syndapin I only in the brain but no ectopic expression in heart or skeletal muscles of *syndapin III* KO mice. 40 µg protein each was loaded per lane. Anti-GAPDH signals served as controls. (**e–h**) Quantitative western blot analyses of homogenates of hearts (**e,f**) and skeletal muscles (**g, h**) from WT and *syndapin III* KO mice addressing putative changes of syndapin II expression levels. Data,

*Figure 2 continued on next page*

*Figure 2 continued*

mean ± SEM. n = 12 each. (**i**) Schematic 3D-view onto a membrane field with caveolae, shallow circular indentations, a longitudinal indentation and orientations of putative random sections (colored) leading to non-representative and often unclear image data calling for views onto wide fields of membrane and 3D-information to ensure more reliable quantitative analyses.

DOI: https://doi.org/10.7554/eLife.29854.006

The following source data is available for figure 2:

**Source data 1.** This spreadsheet contains the data for all quantitative evaluations shown in the different panels of *Figure 2* (i.e. of *Figure 2c,f and h*).

DOI: https://doi.org/10.7554/eLife.29854.007

E-faces, unrelated surfaces (ice areas) in the samples and *syndapin III* KO controls with primary and secondary antibodies evaluated quantitatively (*Figure 3a–f*).

Anti-syndapin III immunolabeling was largely associated with areas of membrane curvature (*Figure 3a,b*). Of all syndapin III immunolabels, 77% were associated with membrane invaginations (*Figure 3c*). The vast majority of them was of caveolar appearance, that is deeply invaginated, circular and uniform in diameter (about 60 nm), with bottoms lighter than the surrounding plasma membrane and dark shadows. The caveolar nature of the deeply invaginated structures was then formally proven by anti-syndapin III and cav3 double-immunogold labeling procedures (*Figure 3d–i'*; *Figure 3—figure supplements 1* and *2*). These analyses thus firmly proved that syndapin III and cav3 colocalized at caveolae of cardiomyocytes.

The immunolabelings of the freeze-fractured samples yielded high anti-cav3 labeling densities of in average 7.1 anti-cav3 labels/deeply invaginated caveola. This ensured the reliable identification of caveolar membrane profiles as caveolar structures independently of their morphology and interestingly unveiled that also circular, shallow indentations showed colocalizations of cav3 and syndapin III. They showed bottoms with the same grey value as the plasma membrane, an only light shadow and an estimated depth of only ~10 nm. Additionally, some cav3 clusters and colocalized syndapin III were also found at completely flat membrane topologies (*Figure 3h',h''*; grey arrows; *Figure 3—figure supplements 1* and *2*).

Quantitative analyses showed that 58% of all observed invaginations were anti-syndapin III-labeled caveolae-like invaginations (deep and shallow) with 35% being colabeled for cav3 and syndapin III (*Figure 3j*).

Strikingly, scoring extended membrane areas (64.8 $\mu m^2$ in total) unveiled that whereas the density of shallow cav3-immunopositive indentations was unchanged, specifically the density of cav3-marked, deep invaginations – that is classical caveolae - was severely reduced in *syndapin III* KO cardiomyocytes when compared to cardiomyocytes prepared from WT mice (*Figure 3i,i',k,l*; *Figure 3—figure supplement 2*). This low density of caveolae observed in cardiomyocytes isolated from *syndapin III* KO mice corresponded well to the reduced caveolae numbers seen in many human caveolinopathies.

In contrast to caveolae, non-caveolar invaginations were rare (*Figure 3j*) and their densities were not statistically significantly changed upon *syndapin III* KO (*Figure 3m*).

Analyses of the distribution of cav3 showed that cav3 at deeply invaginated caveolae decreased in accordance to the reduction of deeply invaginated caveolae upon *syndapin III* KO (*Figure 3n*). Interestingly, however, even with far less abundant caveolae, the cav3 immunolabeling at the plasma membrane still persisted upon *syndapin III* KO (*Figure 3i,i'*; *Figure 3—figure supplement 2*). The overall anti-cav3 immunolabeling density at the plasma membrane of WT and *syndapin III* KO cardiomyocytes remained about the same (*Figure 3o*).

Furthermore, the anti-cav3 immunolabeling also largely remained clustered in *syndapin III* KO cells despite the fact that caveolar invagination was impaired (*Figure 3i,i'*; *Figure 3—figure supplement 2*). The density of the cav3 labeling within ROIs of 150 nm diameter covering caveolae as well as cav3 clusters at shallow or flat membranes also was unchanged in syndapin III KO cells when compared to WT cells (*Figure 3p*).

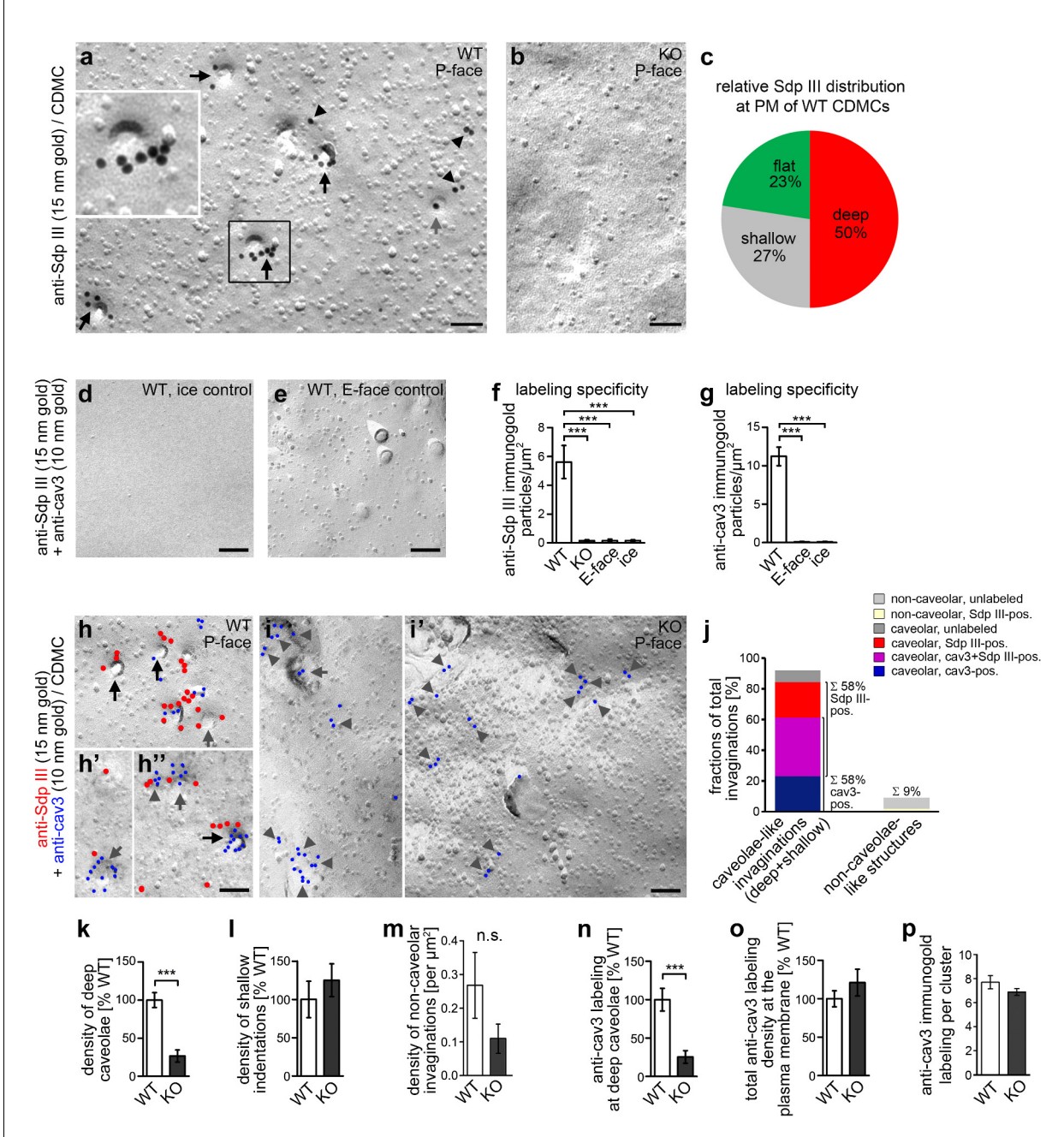

**Figure 3.** *Syndapin III* KO leads to impairments in the formation of cav3-coated caveolae. (**a,b**) Details of wide-field TEM images of anti-syndapin III immunogold-labeled P-faces of freeze-fractured plasma membranes of cardiomyocytes (CDMC) from WT (**a**) and *syndapin III* KO mice (**b**). (**c**) Blinded, quantitative evaluations of the anti-syndapin III labeling distribution on full areas of freeze-fractured membranes. (**a–c**) WT, 15.8 µm$^2$ from 20 images; KO, 27 µm$^2$ from 20 images (two independent cardiomyocyte preparations each pooled from two animals/genotype). (**d-i'**) Electron micrographs of coimmunolabeled control surfaces (**d,e**) and P-faces of WT (**h,h',h''**; for the picture, from which the details in h' and h'' were taken, see *Figure 3—figure supplement 1*) and *syndapin III* KO cells (**i,i'**) as well as blinded quantitative evaluations of labeling densities (**f,g**; n = 20 images each condition) demonstrating the specificity of the labelings at the P-face of WT cardiomyocytes. Syndapin III (15 nm gold, red labels) is present at caveolae highlighted by anti-cav3 labeling (10 nm gold, blue labels). Labelings at caveolae are marked by black arrows, at shallow indentations by grey arrows and at flat membrane areas by grey arrowheads. Bars, 100 nm. For non-color-marked EM micrographs see *Figure 3—figure supplement 2*. (**j**) Analyses of the fractions of caveolae-like profiles (deep and shallow) and of non-caveolar invaginations that were either unlabeled or labeled for cav3, syndapin III and both, respectively (n = 92 WT invaginations). (**k,l**) Blinded quantitative analyses of the relative densities of caveolae (deep, 70 nm in diameter invaginations formally confirmed as caveolae by anti-cav3 labeling) (**k**), and of shallow indentations (**l**), which also were anti-cav3-positive. n = 20 images each; in total, 132 cav3-positive structures were scored. (**m**) Densities of the (rare) non-caveolar invaginations in WT and *syndapin III* KO cardiomyocytes

*Figure 3 continued on next page*

*Figure 3 continued*

(due to the low abundance of such structures (n = 8), n.s.). (n–p) Quantitative analyses of anti-cav3 immunogold labels at deep caveolar invaginations (n; highly significantly decreasing in accordance with the reduced density of (cav3-marked) caveolae), in total (o; n.s) and within cav3 cluster ROIs (150 nm in diameter) (p; n.s.) at the plasma membrane of WT and *syndapin III* KO cardiomyocytes. (j-o) WT, 29.2 µm$^2$ membrane from 20 images; KO, 35.6 µm$^2$ from 20 images (two independent preparations of two animals each). In (p), 107 (KO) and 50 (WT) cav3 cluster ROIs (150 nm diameter,≥4 anti-cav3 immunogold labels) were analyzed. Data, mean ± SEM. Statistical significance, one-way Anova with Tukey's post-test (f,g) and two-tailed Student's t test (k–p), respectively. For further characterization of the primary cardiomyocyte cultures in respect of syndapin III, cav1, cav3 localizations and colocalizations and for phenotypical analyses of cav3 vs. cav1-positive cells see *Figure 3—figure supplement 3*.

DOI: https://doi.org/10.7554/eLife.29854.008

The following source data and figure supplements are available for figure 3:

**Source data 1.** This spreadsheet contains the data for all quantitative evaluations shown in the different panels of *Figure 3* (i.e. of *Figure 3f,g,j–p*).
DOI: https://doi.org/10.7554/eLife.29854.012

**Figure supplement 1.** Syndapin III is present at cav3-labeled caveolae, shallow membrane indentations and flat membrane areas (picture from which *Figure 3h' and h''* were taken).
DOI: https://doi.org/10.7554/eLife.29854.009

**Figure supplement 2.** Syndapin III is present at cav3-labeled caveolae, shallow membrane indentations and flat membrane areas and *syndapin III* KO impairs the formation of caveolae (non-colored data corresponding to images shown in *Figure 3h–h'' and and i,i'*).
DOI: https://doi.org/10.7554/eLife.29854.010

**Figure supplement 3.** Cav1 and cav3/syndapin III show distinct patterns of localization in primary cardiomyocytes, hearts and skeletal muscles with cav3 and syndapin III being restricted to myocytes.
DOI: https://doi.org/10.7554/eLife.29854.011

Taken together, the lack of deep caveolar invaginations in *syndapin III* KO cardiomyocytes was not a consequence of a lack of cav3 at the plasma membrane but strongly suggested a crucial role of syndapin III in the invaginating process of cav3-coated caveolae.

Importantly, *syndapin III* KO specifically resulted in a loss of invaginated cav3-positive caveolae. This specificity for cav3 caveolae coincided with an overlap of syndapin III with cav3 but not with cav1 in immunohistological and EM analyses (*Figure 3—figure supplement 3a,b*). Likewise, cav3 showed no overlap with cav1 (*Figure 3—figure supplement 3c-f*). This argued for a differential expression of cav3 and syndapin III in muscle and cav1 in non-muscle cells. Consistently, cav1-coated caveolae found in the primary cell cultures were unaffected by *syndapin III* KO, that is were invaginated normally (*Figure 3—figure supplement 3f*). It can therefore be firmly concluded that syndapin III is important specifically for invagination of cav3-coated caveolae found in muscle cells.

## Impairment of caveolar invagination upon *syndapin III* KO does not lead to dissociation of CAVIN-1 from the plasma membrane and reveals that cav3 and CAVIN-1 are not sufficient for caveolae formation

CAVIN-1 forms a flexible, net-like protein mesh around caveolin complexes (*Stoeber et al., 2016*) and has been introduced as factor critical for caveolae formation (*Hill et al., 2008*; *Liu and Pilch, 2008*; *Liu et al., 2008*). Thus, we next evaluated whether the striking reduction of caveolar invaginations upon *syndapin III* KO may just be an indirect effect caused by some loss of CAVIN-1 function. Quantitative western blot analyses, however, demonstrated that the expression levels of CAVIN-1 were unchanged in both hearts and skeletal muscles of *syndapin III* KO mice (*Figure 4a,b*). Also, the expression of the other CAVIN proteins (*Cheng et al., 2015*) as well as that of the non-muscle caveolin, cav1, was unchanged (*Figure 4c–f*).

To additionally firmly rule out that a lack of the important coat component CAVIN-1 at specifically the plasma membrane may cause the caveolar impairments observed, we next quantitatively visualized CAVIN-1 by immunolabeling of freeze-fractured *syndapin III* KO and WT cardiomyocytes. In WT cardiomyocytes, CAVIN-1 localized to deeply invaginated, shallowly invaginated and flat membrane areas. As expected, anti-CAVIN-1 immunogold signals were colocalized with the signals of anti-cav3 antibodies, which we used at low concentration to formally prove that the plasma membranes analyzed indeed correspond to cardiomyocytes and not to other cell types (*Figure 4g*; *Figure 4—figure supplement 1*).

Surprisingly, CAVIN-1's membrane association, its ability to form clusters and its colocalization with cav3 seemed not to be impaired upon *syndapin III* KO (*Figure 4h*; *Figure 4—figure*

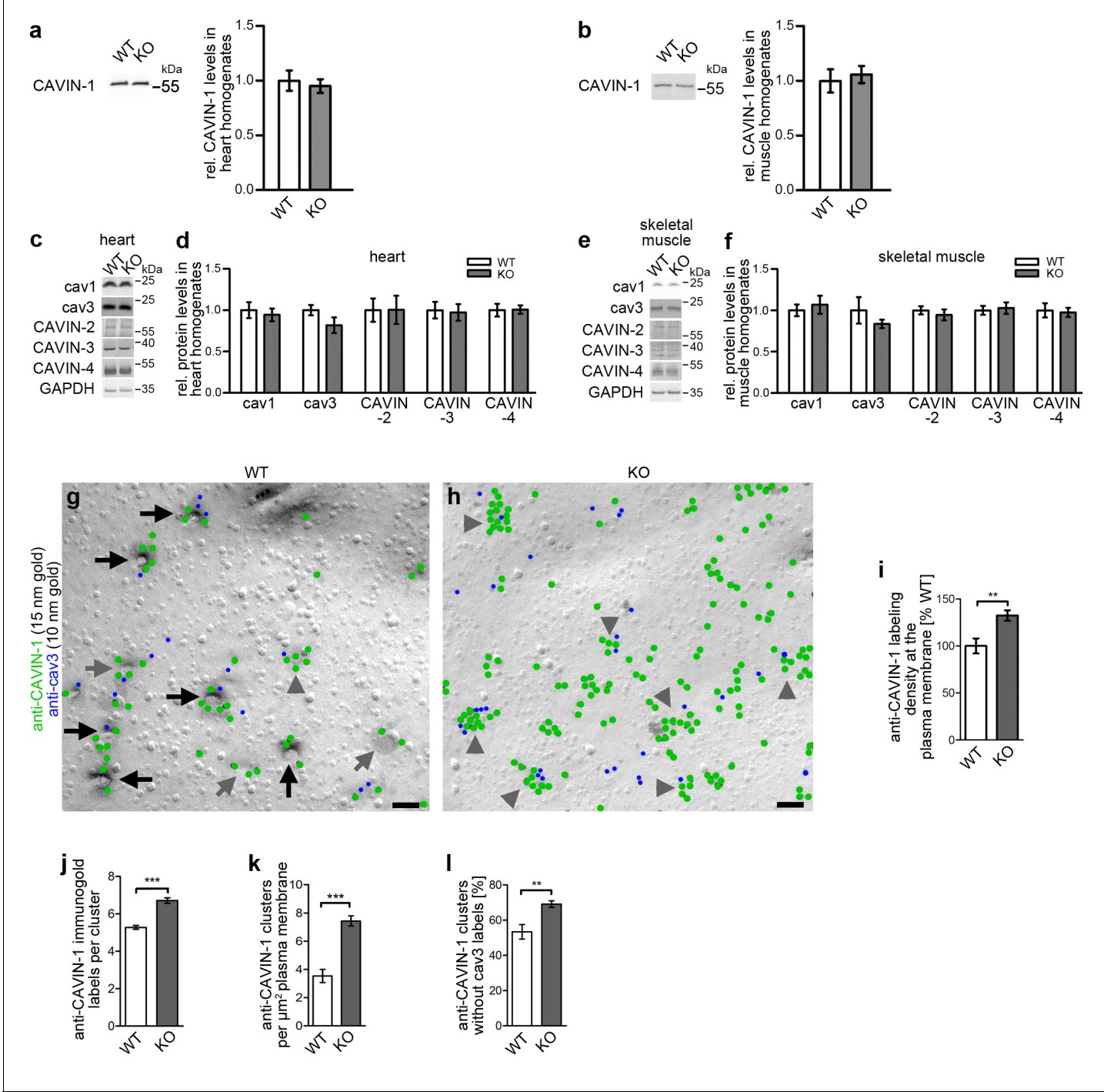

**Figure 4.** Impairment of caveolar invagination by *syndapin III* KO does not lead to dissociation of CAVIN-1 from the plasma membrane. (a,b) Quantitative western blot analyses of homogenates of hearts and skeletal muscles from WT and *syndapin III* KO mice showing that the levels of the important cav3 coat component CAVIN-1 are unaffected. (c-f) Quantitative western blot analyses of homogenates of hearts (c,d) and skeletal muscles (e,f) from WT and *syndapin III* KO mice addressing components suggested to play roles in caveolae formation that may be redundant or related to the critical role of syndapin III in caveolar invagination (normalized to GAPDH). Data, mean ± SEM. N = 12 each genotype. (g,h) Electron micrographs of anti-CAVIN-1 immunogold labeling (green labels) of freeze-fractured cardiomyocytes isolated from WT (g) and *syndapin III* KO mice (h) (in combination with a low concentration immunolabeling of cav3 (blue labels) to prove that indeed a membrane of a (cav3-positive) cardiomyocyte is examined). Note that CAVIN-1's membrane association and its ability to form clusters is not impaired in *syndapin III* KO cells. Examples of clustered labeling of CAVIN-1 at (cav3-positive) deeply invaginated caveolae are marked by black arrows, at shallow indentations by grey arrows and at flat membrane areas by grey arrowheads. For non-color-marked EM micrographs see *Figure 4—figure supplement 1*. Bars, 100 nm. (i,j) Quantitative analyses of the labeling

*Figure 4 continued on next page*

*Figure 4 continued*

densities in WT and *syndapin III* KO samples at the plasma membrane in general (i) and specifically within CAVIN-1 clusters (j). (k,l) Analyses of the density of CAVIN-1 clusters at the plasma membrane (k) and of the percent of clusters without cav3 signal in WT and *syndapin III* KO cardiomyocytes (l). Data, mean ± SEM. Twenty images each (i–l). Statistical analysis, unpaired Student's t test.

DOI: https://doi.org/10.7554/eLife.29854.013

The following source data and figure supplement are available for figure 4:

**Source data 1.** This spreadsheet contains the data for all quantitative evaluations shown in the different panels of *Figure 4* (i.e. of *Figure 4a,b,d,f,i,j,k and l*).

DOI: https://doi.org/10.7554/eLife.29854.015

**Figure supplement 1.** Impairment of caveolar invagination by *syndapin III* KO does not lead to dissociation of CAVIN-1 from the plasma membrane (non-colored images corresponding to *Figure 4g and h* and further example images).

DOI: https://doi.org/10.7554/eLife.29854.014

*supplement 1*). These results were unexpected, as CAVIN-1 was suggested to dissociate from flattened membranes, as evaluated by immunofluorescence analyses of clusters of overexpressed CAVIN-1-mCherry fusion proteins in HeLa cells subjected to hypoosmotic shock (*Sinha et al., 2011*).

However, quantitative evaluations of our ultra-high-resolution imaging experiments clearly confirmed that CAVIN-1 levels at the plasma membrane were not reduced upon *syndapin III* KO and/or the presence of predominantly flat cav3 clusters caused by the loss of syndapin III. Instead, the CAVIN-1 density at the plasma membrane even moderately increased by about 30% (*Figure 4i*). The CAVIN-1 density within CAVIN-1 clusters rose accordingly (*Figure 4j*). Since also the density of CAVIN-1 clusters found was increased (*Figure 4k*), these increases of CAVIN-1 in clusters at *syndapin III* KO plasma membranes of cardiomyocytes obviously came at the expense of the disperse labeling. Many of these additional CAVIN-1 clusters at the plasma membrane, however, were negative for cav3 (*Figure 4l*).

Taken together, all of our examinations thus strongly suggest that the loss of cav3-caveolae can be specifically attributed to syndapin III loss-of-function. Our observations also unveiled that even a combination of cav3 and CAVIN-1 residing at the plasma membrane is insufficient for caveolar invagination, if syndapin III is lacking.

## Syndapin III shapes liposomes into tubules with caveolar diameters and localizes to the rim of cav3 coats

To shed light on the mechanisms, by which syndapin III may shape the plasma membrane into caveolar invaginations, two important questions needed to be addressed: Does syndapin III have the ability to generate membrane curvatures that resemble the 70 nm diameters of caveolae? And if so, would syndapin III integrate everywhere into the cav3 coat to control its topology or rather act as a locally restricted curvature-inducer? The first question seemed critical, as in vitro-reconstitutions with liposomes analyzed by negative-staining procedures suggested that a fragment of syndapin III can induce membrane curvatures but the diameters were very invariant (around 110 nm [*Bai et al., 2012*]), that is they did not fit caveolar diameters. However, our analyses with full-length syndapin III using two different methods of evaluation, freeze-fracture/TEM as well as cryoTEM, showed that syndapin III had mechanistic properties in line with its caveolar localization and function. Both tagged and untagged full-length syndapin III formed tubules with an average diameter of about 80 nm (*Figure 5a–e*). The observed diameter range of 75–90 nm corresponded very well to the membrane topologies that were immuno-positive for syndapin III in vivo. Cav3-caveolae had diameters of 60–80 nm (*Figure 3*).

The fact that syndapin III seemed to prefer curvatures that are less than those observed for established cav3 coats suggested that syndapin III may not be an integral part of invaginated cav3 coats but may reside at areas that are about 10 nm wider. EM tomograms and 3D-segmentations of membrane topology in combination with the 3D-detection of immunogold labels indeed demonstrated that syndapin III was not found at the lateral walls or at the bottom of caveolar invaginations but was restricted to areas of membrane curvature transition from the caveolar invagination to the continuum of the plasma membrane (*Figure 5f–h*; *Videos 1* and *2*).

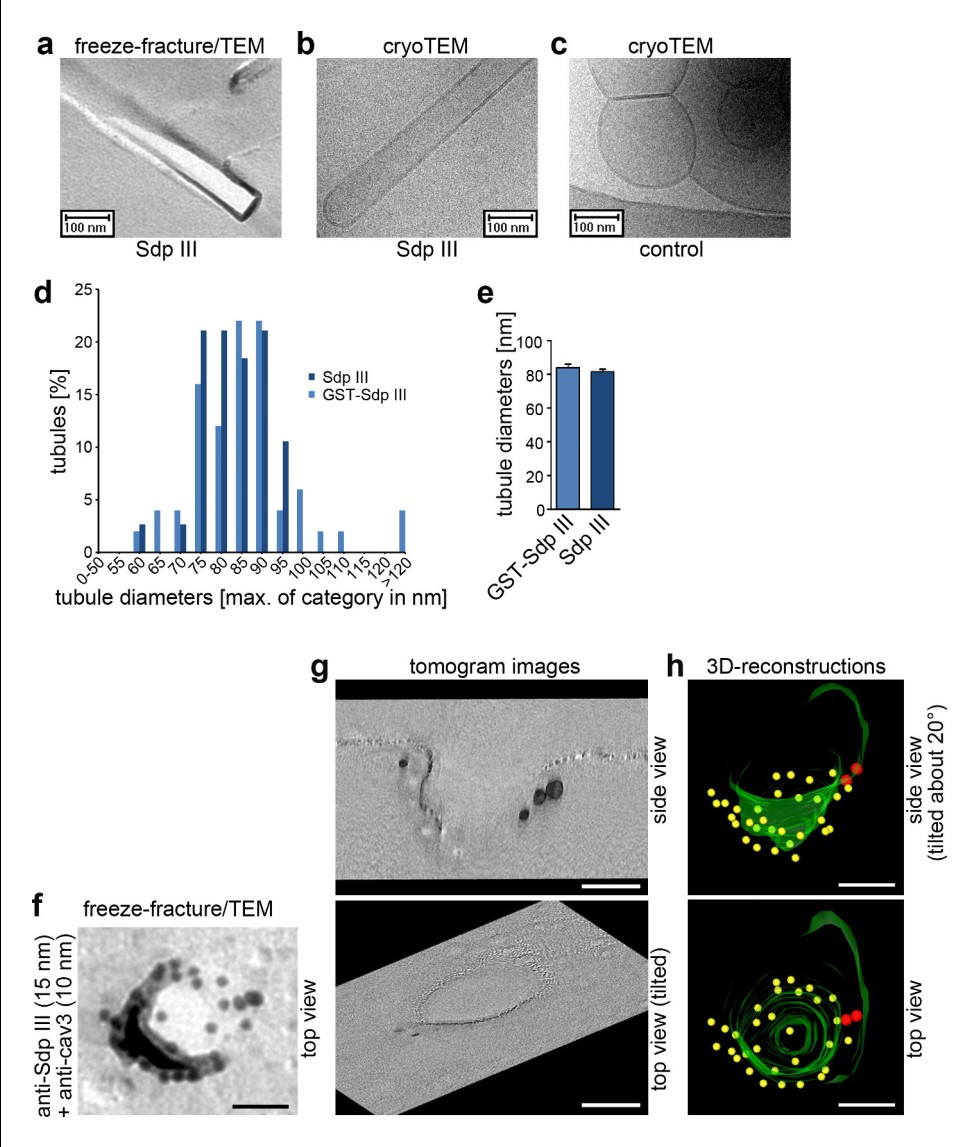

**Figure 5.** Syndapin III shapes liposomes into tubules with caveolar diameters and localizes to the rim of cav3 coats. (a–c) Analyses of tubules induced by incubating liposomes with syndapin III (a,b) and GST (c), respectively, by freeze-fracturing/TEM (a) as well as by cryo-TEM (b,c). (d,e) Quantitative analyses of tubule diameter distributions (d) and averages of diameters (e) induced by syndapin III and GST-syndapin III, respectively. Data in (e), mean +SEM. (d,e) n= 50 (GST-syndapin III) and n = 38 (syndapin III) freeze-fractured tubuli. (f) High resolution, 80 kV top view of a deeply invaginated caveolar structure of a WT cardiomyocyte immunolabeled for syndapin III (15 nm gold) and cav3 (10 nm gold). Bar, 50 nm. (g,h) Corresponding 120 kV tomogram images (g) and views from a 3D-reconstruction (h) show that syndapin III is at the edge of the cav3 coat (34 caveolae analyzed; two full 3D-segmentations; invaginated membrane, green; immunogold labels cav3, yellow; syndapin III, red). Bars in f–h), 50 nm.

DOI: https://doi.org/10.7554/eLife.29854.016

The following source data is available for figure 5:

**Source data 1.** This spreadsheet contains the data for all quantitative evaluations shown in the different panels of *Figure 5* (i.e. of *Figure 5d and e*).

DOI: https://doi.org/10.7554/eLife.29854.017

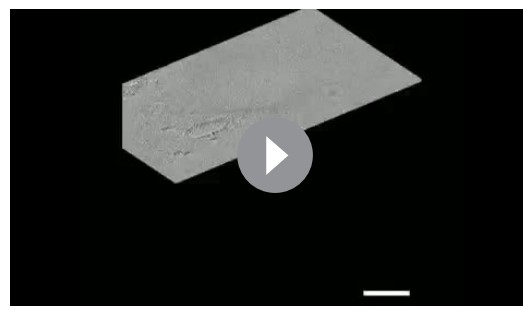

**Video 1.** Electron tomogram data set of the caveolar invagination shown in *Figure 5f,g*. Cav3 and syndapin III are indicated by the electron-dense gold particles (anti-syndapin III, 15 nm; anti-cav3, 10 nm). Reconstruction and tomographic sectioning carried out with IMOD software. Bar, 50 nm.
DOI: https://doi.org/10.7554/eLife.29854.018

## Syndapin III organizes cav3-coated caveolae and cav3-associated lipid domains

Coprecipitations from heart lysates confirmed that the syndapin III F-BAR-domain interacts with cav3 coats. Interestingly, similar to the caveolin coat component CAVIN-1 showing only weak – if any – interactions with caveolin (*Hill et al., 2008*; *Liu and Pilch, 2008*; *Mohan et al., 2015*), the interactions were rather weak and therefore suggested that syndapin III may not represent a general and integral part of cav3 coats (*Figure 6a*).

However, similar to syndapin II and cav1 in some cell lines (*Koch et al., 2012*; *Senju et al., 2011*), overexpression of the F-BAR-domain of syndapin III led to colocalization of endogenous cav3 at F-BAR-enriched sites at the plasma membrane of primary cardiomyocytes (*Figure 6b,c*). Syndapin III may thus modulate membrane topologies and thereby organize cav3 coats at muscle cell membranes.

Caveolae are associated with cholesterol-enriched lipid domains (*Harder et al., 1997*). Preparations of TritonX-100-resistant membranes (DRMs) demonstrated that cav3 and syndapin III both floated together (*Figure 6d,e*).

Interestingly, cav3 showed an altered distribution upon *syndapin III KO* (*Figure 6f–i*; *Figure 6—figure supplement 1*). In WT heart samples, syndapin III and cav3 floated to fraction F4+F5. *Syndapin III* KO fractions F4+F5 were devoid of cav3 but cav3 mainly floated to fractions F6+F7 (*Figure 6d,f,h*; *Figure 6—figure supplement 1a,b*). Related shifts of cav3 toward higher densities were observed for preparations from *syndapin III* KO skeletal muscles (*Figure 6e,g,i*; *Figure 6—figure supplement 1c,d*). These anti-cav3-immunopositive density gradient positions of *syndapin III* KO DRMs still were distinct from all other cellular compartments tested (*Figure 6j,k*).

Together, these data strongly suggest that syndapin III plays a critical role in organizing caveolar coats and associated lipid environments.

## Dissecting the functions of caveolae from those of cav3

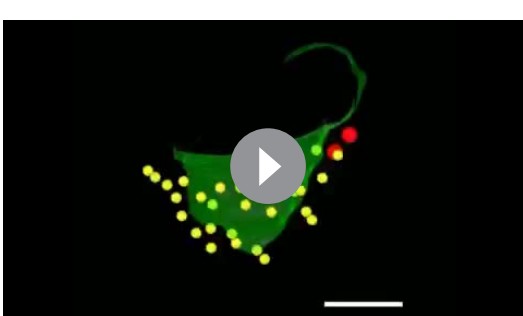

**Video 2.** Rotation of the 3D segmentation of a syndapin III and cav3-coated caveolar invagination shown in *Figure 5h*. Syndapin III is indicated by the red spheres and cav3 is indicated by the yellow spheres. The Pt/C layer of freeze-fractured cardiomyocyte plasma membrane is indicated (green). Segmentation carried out with IMOD software. Bar, 50 nm.
DOI: https://doi.org/10.7554/eLife.29854.019

*Caveolin* KO analyses suggested that caveolae represent crucial signaling hubs (*Hansen and Nichols, 2010*). The physiological defects in *caveolin*-deficient mice may thus in part relate to the observed ERK1/2 hyperactivation rather than to the cellular functions of caveolar invaginations. Furthermore, *cav3* KO was reported to impair lipid raft associations of the dystrophin-glycoprotein complex and to result in its loss (*Woodman et al., 2002*).

We therefore vigorously tested *syndapin III* KO mice for these phenotypes. Relative phosphoERK1/2 levels were not altered upon *syndapin III* KO (*Figure 7a,b*).

Dystrophin impairments were not observed either. No localization defects of dystrophin and β-dystroglycan could be detected in sections of skeletal muscles from *syndapin III* KO mice (*Figure 7c–f*). Also dystrophin and β-dystroglycan expression levels were unaltered upon *syndapin*

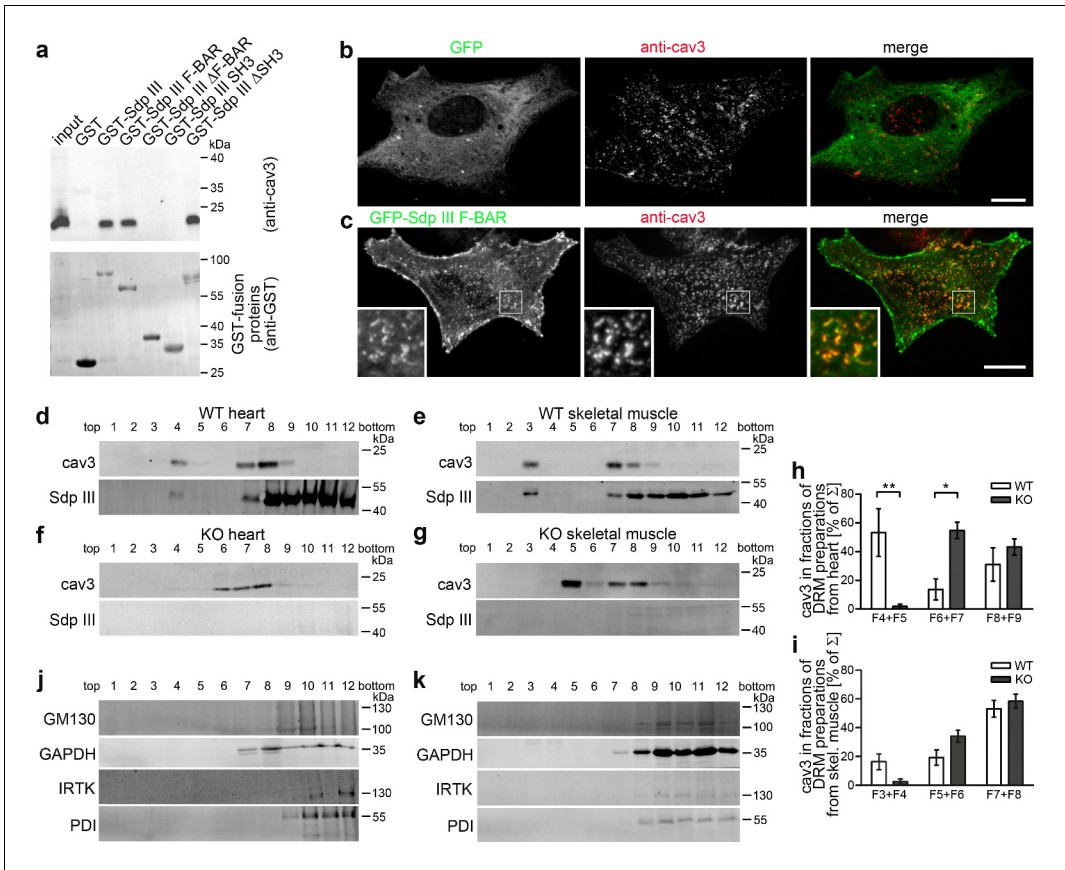

**Figure 6.** Syndapin III is involved in the organization of cav3-containing membrane domains. (a) Coprecipitation of endogenous cav3 from heart lysates with the indicated immobilized GST-fusion proteins of syndapin III. (b,c) GFP-syndapin III F-BAR domain but not GFP clusters with cav3 at the membrane of primary cardiomyocytes. Boxed areas, higher magnification insets. Bars, 10 μm. (d–k) *Syndapin III* KO changes the biophysical properties of cav3-containing DRMs. Immunoblottings of cav3 and syndapin III (d–g) and proteins representing Golgi (GM130), cytosol (GAPDH), plasma membrane (IRTK) and ER (PDI) (j,k) in TritonX-100-resistant membrane preparations from heart and skeletal muscles from WT (d,e) and *syndapin III* KO mice (f,g,j,k). Quantitative analyses (h,i) demonstrate the shift of cav3-containing TritonX-100-resistant membranes from fractions F4 +F5 to F6 +F7 in heart (d,f,h) and from F3 +F4 to F5 +F6 in skeletal muscle (e,g,i) upon *syndapin III* KO. Hearts, n = 3 each genotype; skeletal muscles, n = 9 each genotype. Data, mean ± SEM. Statistical significances, two-way ANOVA and Bonferroni post-test. For individual comparisons of WT and KO fractions see *Figure 6—figure supplement 1*.

DOI: https://doi.org/10.7554/eLife.29854.020

The following source data and figure supplement are available for figure 6:

**Source data 1.** This spreadsheet contains the data for all quantitative evaluations shown in the different panels of *Figure 6* and of the corresponding *Figure 6—figure supplement 1* (i.e. of *Figure 6h and i* and *Figure 6—figure supplement 1a–d*).
DOI: https://doi.org/10.7554/eLife.29854.022

**Figure supplement 1.** Syndapin III plays an important role in organizing cav3-containing membrane domains.
DOI: https://doi.org/10.7554/eLife.29854.021

*III* KO in both heart and skeletal muscle (*Figure 7g–j*).

## Syndapin III-mediated caveolar invagination counterpoises mechanical stress and thereby ensures cellular integrity

Increases of membrane tension, as mimicked by applying hypoosmotic stress, led to fewer GFP-cav1-coated caveolae in HeLa and MLEC cells (*Sinha et al., 2011*). Our TEM analyses of freeze-fractured NIH3T3 cells, which we labeled for cav1 (in average 13.9 anti-cav1 labels/caveola (iso)), confirm that the immediate cell response to increased membrane tensions induced by hypoosmotic conditions is a reduction of caveolar invaginations (*Figure 8—figure supplement 1a–d*). In NIH3T3 cells,

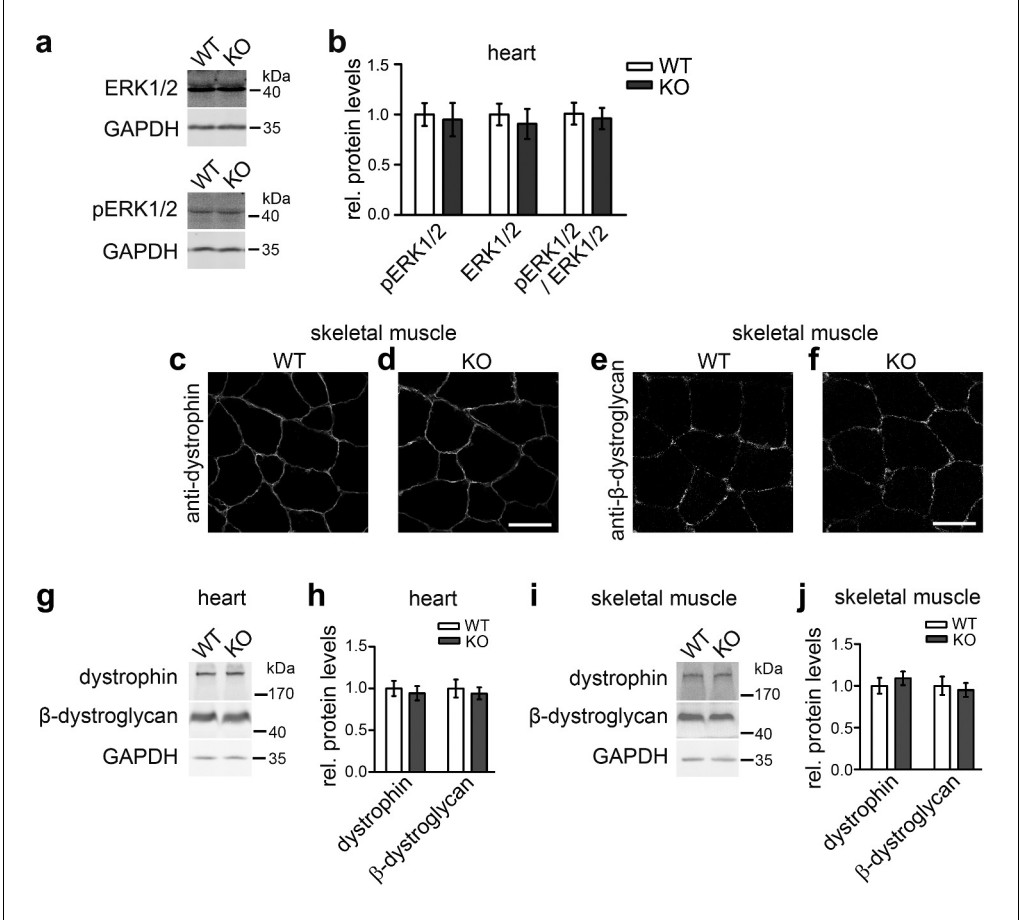

**Figure 7.** Not all described cav3 loss-of-function phenotypes are found upon *syndapin III* KO and may thus not reflect impairments of caveolar invagination. (**a,b**) Quantitative immunoblot analyses of ERK1/2 and phosphoERK1/2 in heart homogenates of WT and *syndapin III* KO mice (normalized to WT) show no alteration of MAPK signaling (pERK1/2/ERK1/2 signals). Data represent mean ± SEM; n = 12 each. (**c–f**) Unchanged subcellular localization of both dystrophin and β-dystroglycan in sections of skeletal muscles from *syndapin III* KO mice. Bars, 50 μm. (**g–j**) Levels of dystrophin and β-dystroglycan are unchanged in heart and skeletal muscle homogenates upon *syndapin III* KO (normalized to WT). Data represent mean ± SEM; n = 12 each.
DOI: https://doi.org/10.7554/eLife.29854.023

The following source data is available for figure 7:

**Source data 1.** This spreadsheet contains the data for all quantitative evaluations shown in the different panels of *Figure 7* (i.e. of *Figure 7b,h and j*).
DOI: https://doi.org/10.7554/eLife.29854.024

cav1 hereby remained at the plasma membrane (*Figure 8—figure supplement 1e–h*). These data strongly support a role of caveolae as mechanosensors buffering mechanical stress.

The observed counteractions do not just reflect abilities of cav1-expressing transformed, immortal cell lines, such as HeLa, MLEC (*Sinha et al., 2011*) or NIH3T3 (this study) but primary mouse cardiomyocytes showed a similar behavior. The induced mechanical stress dramatically reduced deeply invaginated, cav3-positive structures (−75%) (*Figure 8a–d*). Again, the anti-cav3 labeling density at the plasma membrane remained constant upon hypoosmotic flattening of caveolae (*Figure 8e*). The distribution of cav3 changed in accordance with the flattened membrane topology (*Figure 8f–h*).

A function of caveolae in buffering mechanical stress leads to the prediction that a lack of invaginated caveolae will render myocytes vulnerable to abrupt changes of membrane tensions. To test this hypothesis experimentally in primary cells, we compared the structural integrity of WT and *syndapin III KO* cardiomyocytes under such conditions using Trypan Blue (*Figure 8i–o*). WT cells were

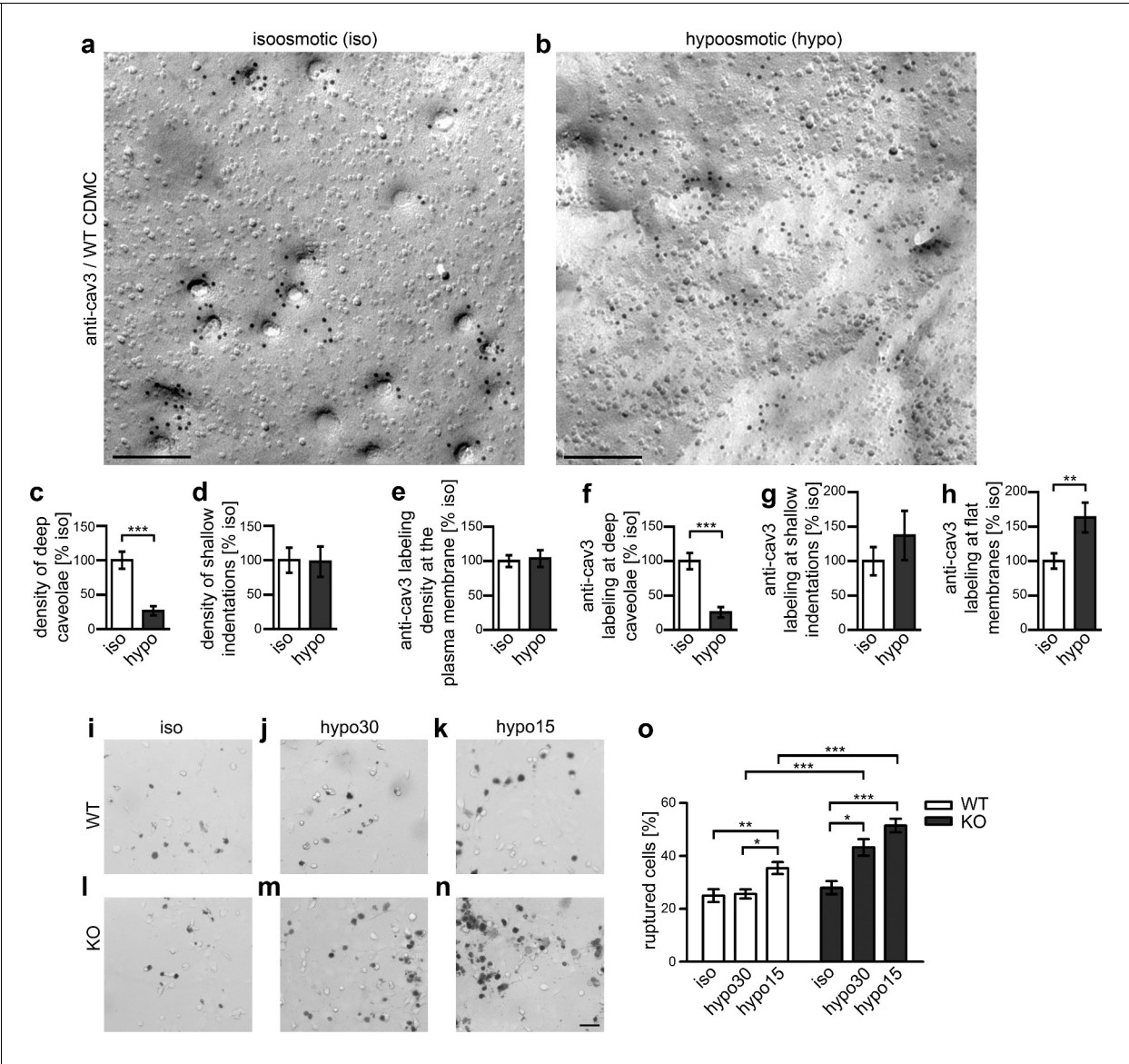

**Figure 8.** Syndapin III-mediated caveolar invagination counterpoises membrane tensions and thereby ensures cell integrity. (a,b) Anti-cav3-labeled P-faces of freeze-fractured plasma membranes of cardiomyocytes from WT mice incubated for 5 min in isoosmotic (iso) (a) and hypoosmotic conditions (hypo; hypo15 buffer) (b). Bars, 200 nm. (c,d) A dramatic reduction of deeply invaginated caveolar structures was observed upon the induced, cell swelling-mediated rise in membrane tension caused by hypo conditions. (e–h) Quantitation of the anti-cav3 labeling distribution in total and in relations to deeply invaginated, shallow and flat membrane topologies (relative to isoosmotic condition). Data (c–h), mean ± SEM. iso, 148.8 $\mu m^2$ membrane area from 43 images, 251 caveolar invaginations; hypo, 124.6 $\mu m^2$ membrane area from 36 images, 66 caveolar invaginations, two independent cardiomyocyte preparations and assays. Statistical significance, two-tailed Student's t test. For related examinations of caveolar flattening upon induction of membrane tensions in NIH3T3 cells please see *Figure 8—figure supplement 1*. (i–o) Trypan Blue assays with WT (i–k) and *syndapin III* KO cardiomyocytes (l–n) subjected to membrane tensions mimicked by mild hypoosmotic stress (5 min hypo30; j,m) and stronger hypoosmotic stress (5 min hypo15; k,n) unveiling a higher vulnerability of *syndapin III* KO cells under conditions increasing membrane tensions. Note the increased abundancy of Trypan Blue-positive, ruptured cardiomyocytes upon hypoosmotic stress. Bar in n (for i–n), 50 $\mu m$. (o) Quantitative data, mean ± SEM of 18 blinded experiments (about 100 cells each assay) from six independent preparations of cardiomyocytes/genotype. One-way Anova with Tukey's post-test.

DOI: https://doi.org/10.7554/eLife.29854.025

The following source data and figure supplement are available for figure 8:

**Source data 1.** This spreadsheet contains the data for all quantitative evaluations shown in the different panels of *Figure 8* and of the corresponding *Figure 8—figure supplement 1* (i.e. of *Figure 8c–h* and of the *Figure 8—figure supplement 1c–h*).

DOI: https://doi.org/10.7554/eLife.29854.027

**Figure supplement 1.** Caveolar flattening upon induction of membrane tensions in NIH3T3 cells.

*Figure 8 continued on next page*

*Figure 8 continued*

DOI: https://doi.org/10.7554/eLife.29854.026

able to fully cope with moderate stress (hypo30) and were only moderately affected by stronger hypoosmotic stress (hypo15) (*Figure 8i–k,o*). In contrast, *syndapin III KO* cardiomyocytes showed a significantly higher vulnerability under both conditions (*Figure 8l–o*).

## Severe impairment of caveolar invagination has no obvious cardiac consequences

Interestingly, the inability of *syndapin III* KO mice to invaginate caveolae had no obvious consequences for heart function and integrity. *Syndapin III KO* mice were viable and heart sections of juvenile mice did not show any hypertrophic cardiac myocytes (*Figure 9a–c*). Heart weights (data not shown separately) and heart/body weight ratios were unaltered (*Figure 9d,e*). Left ventricle wall thickness also did not differ from WT (*Figure 9f–h*)

Even at >64 weeks of age, sections of *syndapin III* KO hearts did not show signs of cardiomyocyte hypertrophy or necrosis. Physical exercise-induced necrosis was not detectable either (*Figure 9i–l*).

These data raised the questions whether i) the impairment of caveolar invagination upon *syndapin III KO* merely is a phenomenon of dissociated cells or whether ii) caveolar compensation of membrane tensions, as mimicked by hypoosmotic stress in cell lines (*Sinha et al., 2011*) (*Figure 8—figure supplement 1*) and in primary cardiomyocytes (*Figure 8a–h*), is physiologically irrelevant. It was therefore required to confirm or dismiss impairments of caveolar invagination upon *syndapin III* KO at the tissue level.

In line with our data in primary cardiomyocytes (*Figures 2* and *3*), we observed almost no caveolar profiles in sections of fixed *syndapin III* KO tissue subjected to TEM (our unpublished results). To again circumvent the technical caveats of quantitative analyses using this conventional method, we established freeze-fracturing and immunolabeling procedures for muscle tissues. Our analyses of freeze-fractured and immunolabeled *syndapin III* KO hearts clearly demonstrated that, also in the tissue context, caveolar invagination was severely impaired (*Figure 9m,n*). Quantitative analyses showed that the effects even were slightly stronger than in dissociated cells. Upon *syndapin III* KO, only 11.7% of the caveolae found in WT remained. Even the density of the shallow, cav3-positive indentations was significantly decreased in *syndapin III* KO hearts (*Figure 9o,p*).

Consistent with the observations in dissociated cells, the anti-cav3 labeling density at membranes of freeze-fractured *syndapin III* KO hearts in total remained unchanged (*Figure 9q*). The unchanged presence of cav3 at the membrane was in line with unchanged cav3 levels in TritonX-100-resistant membrane fractions from heart observed by quantitative western blotting (*Figure 9r*). In accordance with the reduced densities of caveolae and cav3-positive indentations, cav3 labeling at deep and shallowly invaginated caveolar structures was strongly decreased (*Figure 9s,t*).

We thus concluded that the *syndapin III* KO-mediated impairment of caveolar invagination is not limited to isolated, cultured cells but is also valid in hearts of *syndapin III* KO mice but not critical for this organ.

## Impairments in caveolar invagination lead to human disease-related skeletal muscle defects

LGMD-1C, distal myopathy and RMD are degenerative, skeletal muscle-associated diseases associated with mutations in *CAV3* (*Gazzerro et al., 2010*). Skeletal muscles subjected to freeze-fracturing and immunogold labeling procedures clearly showed that specific anti-syndapin III immunolabeling is present at caveolae of skeletal muscles (*Figure 10a*; compare *Figure 10b*). A significant portion of the caveolae found in muscle tissues were immunopositive for syndapin III (*Figure 10c*). Non-caveolar invaginations were very rare and their density did not show any significant change upon *syndapin III* KO (*Figure 10c,d*).

The presence of syndapin III at caveolae was corroborated by confocal microscopy analyses of longitudinal sections of skeletal muscle. Syndapin III clearly colocalized with cav3 puncta of different sizes and the Pearson correlation coefficient was clearly positive (*Figure 10e–g*).

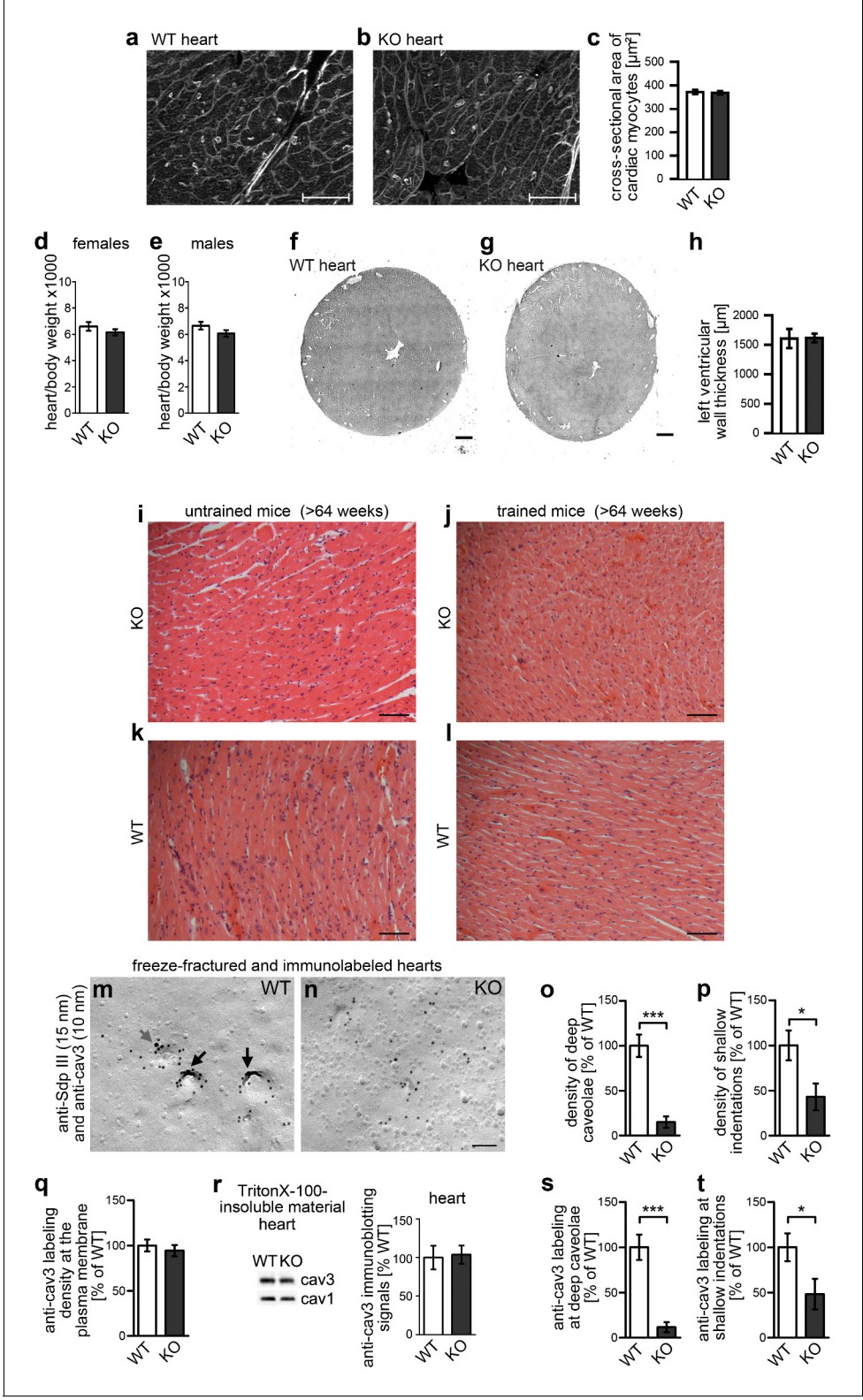

**Figure 9.** Impairment of caveolar invagination upon *syndapin III* KO in heart tissue has no consequences on cardiac integrity. (a–c) Wheat germ agglutinin stainings of 8 μm cryosections of hearts of 20 weeks old *syndapin III* KO mice (b) show no signs of cellular damage or alteration of cross-sectional areas of cardiac myocytes when compared to WT (a). Bars, 50 μm. Quantitative data (c) represent mean ± SEM. WT, 273 cells, 6 animals; KO, 378 cells, 7 animals. Statistical significance was tested using two-tailed Student's t test. p=0.7462 (n.s.). (d,e) Ratios (x1000) of heart and body weights of about 4-month-old female (d) and male (e) mice. WT, eight female and male mice each; KO, seven female and nine male mice. Statistical significance,

*Figure 9 continued on next page*

*Figure 9 continued*

two-tailed Student's t test. Female mice, p=0.3132 (n.s.); male mice, p=0.1377 (n.s.). (**f–h**) H&E stainings of WT and *syndapin III* KO heart cryosections show no left ventricular wall thickening upon *syndapin III* KO. Bars, 500 µm. Quantitative data represent mean ± SEM. WT, seven animals; KO, nine animals. Statistical significance, two-tailed Student's t test. p=0.9451 (n.s.). (**i–l**) H&E stainings of paraffin sections of aged WT and *syndapin III* KO myocard (three animals each) display no signs of cellular damage or of necrosis irrespective of whether mice were trained or not. Bar, 50 µm. (**m,n**) Details of wide-field views of P-faces of freeze-fractured heart tissues double-immunogold labeled for cav3 and syndapin III. Note that, also in heart tissue, the formation of caveolae is impaired upon *syndapin III* KO. In WT, syndapin III and cav3-immunopositive caveolae are marked (long black arrows; shallow, cav3-positive indentation, short grey arrow). Bar, 100 nm. (**o,p**) Quantitative determinations of caveolae (**o**) and of shallow, cav3-positive membrane indentations (**p**) in WT and *syndapin III* KO hearts. (**q**) Quantitative assessment of the anti-cav3 labeling densities at membranes of freeze-fractured *syndapin III* KO hearts relative to WT (n.s.). (**r**) Quantitative western blotting of TritonX-100-resistant membrane fractions from WT and *syndapin III* KO hearts showing unchanged cav3 levels. 25 µg protein loaded each. n = 6 each genotype. (**s,t**) Quantitative assessment of the cav3 immunolabeling at caveolae (**s**) and shallow indentations (**t**). Note that the observed decreases are in accordance with the reduced abundance of these structures (compare **s,t**) vs. (**o,p**). (**o–q, s,t**) WT, 72.7 µm² membrane from 21 images; KO, 72.7 µm² membrane from 21 images from two independent cardiomyocyte preparations (from two hearts) each genotype. In total, 107 cav3-positive structures scored. Data, mean ± SEM. Statistical significance, two-tailed Student's t test.

DOI: https://doi.org/10.7554/eLife.29854.028

The following source data is available for figure 9:

**Source data 1.** This spreadsheet contains the data for all quantitative evaluations shown in the different panels of *Figure 9* (i.e. of *Figure 9c–e,h and o–t*).

DOI: https://doi.org/10.7554/eLife.29854.029

Quantitative analyses of anti-cav3 immunofluorescence signals at the plasma membrane, intracellularly and in the cav3 puncta at the plasma membrane showed no differences between WT and *syndapin III* KO muscles (*Figure 10h–j*). Quantitative western blotting of TritonX-100-resistant membrane fractions from skeletal muscle were in line with the quantitative visual examinations and showed no difference between genotypes (*Figure 10k*). Also quantitative evaluations of EM images recorded from anti-cav3 immunostained freeze-fractured muscles showed that the amount of cav3 at the plasma membrane was not reduced upon *syndapin III* KO (*Figure 10l*).

3D rendering of confocal image stacks of longitudinally cut muscle fibers showed that 32% of all cav3 spheres were positive for syndapin III (*Figure 10m,n*). These data are somewhat reminiscent of an estimation of the syndapin II isoform (Pacsin2) being colocalized with about a third of cav1 clusters in HeLa cells (*Hansen et al., 2011*).

Yet, despite the fact that syndapin III is only found at a (albeit significant) fraction of caveolae at a given time and despite the undisturbed presence of cav3 at the plasma membrane, the density of deeply invaginated caveolae was dramatically reduced upon *syndapin III* KO (*Figure 10o*). Also the shallow, cav3-positive indentations were strongly reduced albeit not as dramatically as the deeply invaginated caveolae (*Figure 10o,p*).

The distribution of cav3 changed accordingly. The amount of cav3 at deep and shallow cav3-marked caveolar invaginations was strongly reduced (*Figure 10q,r*). Interestingly, cav3 clusters still persisted, that is cav3 did not largely disperse despite the flat membrane topologies and the lack of caveolae in *syndapin III* KO skeletal muscle fibers. Instead, freeze-fractured skeletal muscles from *syndapin III* KO mice still showed about 70% of the cav3 cluster density observed in WT muscles (*Figure 10s*) and the cav3 density within the persisting cav3 cluster ROIs only showed a minor reduction (−7%; *Figure 10t*). Taken together, syndapin III KO thus seems to moderately impair cav3 clustering but does not affect the cav3 abundance within the clusters that are formed and does not interfere with the presence of cav3 at the plasma membrane in general (*Figure 10l*).

Surprisingly, despite a relatively severe loss of deeply invaginated caveolae upon *syndapin III KO*, detailed pathomorphological examinations using for example acidic phosphatase, periodic acid-Schiff, Congo Red, ATPase, iron, lactate dehydrogenase and hematoxylin and eosin (H&E) stainings did not unveil any dramatic defects of *syndapin III* KO muscles (H&E, *Figure 11a,b*).

H&E staining of sections of WT and *syndapin III* KO muscle fibers, however, revealed a sign of necrosis in one section from *syndapin III* KO muscles (our unpublished data). This minor indication for muscular defects in *syndapin III* KO mice and the fact that also some of the human CAV3-associated myopathies show relatively mild symptoms and late onset (*Gazzerro et al., 2010*), respectively, prompted us to next subject aged mice to physical exercises. Strikingly, we observed genotype-

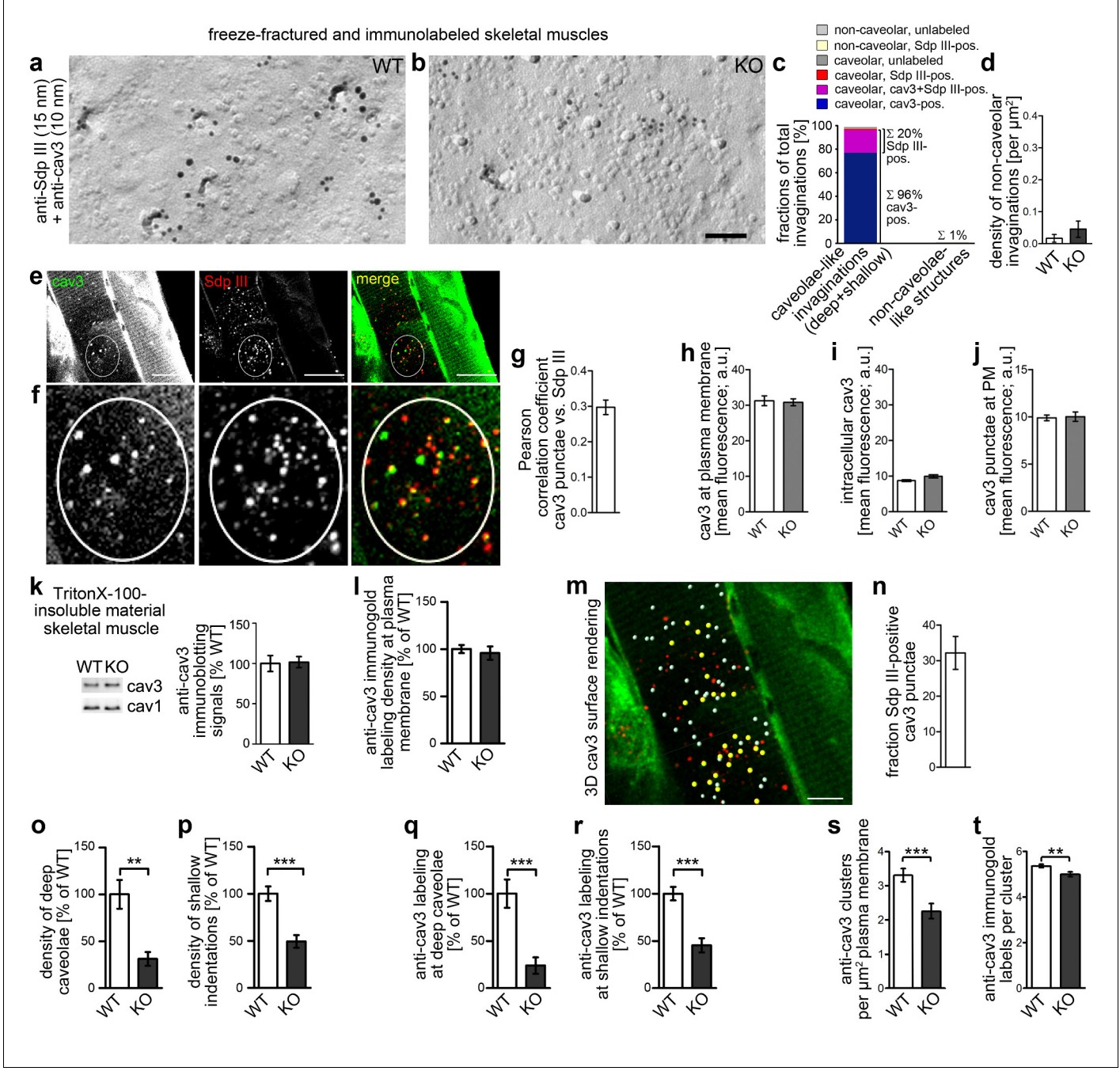

**Figure 10.** *Syndapin III* KO causes impairments in caveolar invagination in skeletal muscle. (a,b) Double-immunogold labeled, freeze-fractured skeletal muscles of WT and *syndapin III* KO mice (syndapin III, 15 nm gold; cav3, 10 nm gold) show that caveolar invagination is impaired in tissues of *syndapin III* KO mice. Bar, 200 nm. (c) Analyses of the fractions of caveolar-like profiles (deep and shallow) and non-caveolar invaginations labeled for cav3, syndapin III and both in freeze-fractured muscles. In total, 291 invaginations from WT skeletal muscles were evaluated. (d) Densities of the (extremely rare) non-caveolar invaginations in WT and *syndapin III* KO cardiomyocytes (low abundance of such structures, in total only n = 2 (WT) and n = 3 (KO) found; n.s.). (e–j) Colocalization analyses based on confocal images of longitudinal sections of skeletal muscles immunostained for cav3 and syndapin III. (e,f) Example images. Encircled is a ROI for colocalization analyses at puncta (e); (f) shows magnification and (g) shows the corresponding Pearson correlation coefficient (n = 60 ROIs). (h–j) Quantitative analyses of anti-cav3 immunofluorescence signals at the plasma membrane ((h); n = 230 ROIs each genotype in transversal sections), in intracellular volumes ((i); n = 160 ROIs each genotype in transversal sections) and in cav3 puncta ((j); n = 60 each genotype in longitudinal sections). (k) Quantitative immunoblotting of TritonX-100-insoluble material showed no differences in cav3 and cav1 levels upon *syndapin III* KO. 25 μg protein loaded each. WT, 12 animals; KO, 12 animals. (l) Quantitative analyses of immunogold-labeled, freeze-fractured skeletal muscle fibers showing no difference in anti-cav3 labeling density between WT and *syndapin III* KO muscles at the plasma membrane. (m,n) Example of 3D surface rendering (using IMARIS) of confocal image stacks of the anti-cav3 immunolabeling shown in overview (e) for determination

*Figure 10 continued on next page*

*Figure 10 continued*

of the frequency of syndapin III-positive cav3 puncta. Yellow spheres, syndapin III-positive; white spheres, syndapin III-negative cav3 3D surface rendered puncta. (n) Quantitative analyses of the frequencies of spatial overlap of syndapin III with cav3 at the level of light microscopy (14 ROIs). (o,p) Quantitation of the densities of deep caveolae and shallow indentations in electron micrographs of freeze-fractured skeletal muscles show severe impairments of caveolar invagination in vivo. (q,r) The relative density (WT = 100%) of the anti-cav3 labeling associated with invaginated, caveolar membrane topologies of *syndapin III* KO muscle membranes changes in accordance with the reduced densities of invaginated caveolar structures. WT, 128 $\mu m^2$ membrane from 37 images; KO, 69.2 $\mu m^2$ membrane from 20 images from two muscle preparations. In total, 400 cav3-positive structures scored. (s,t) Analyses of the density of cav3 clusters at the plasma membrane of freeze-fractured skeletal muscle fibers (s) and of the anti-cav3 immunogold labeling density in cluster ROIs (t). Analyses in (s) were done by image and expressed as clusters/$\mu m^2$ (*Nassoy and Lamaze, 2012*). n = 20 images each genotype. (t) n = 377 ROIs (WT) and 148 ROIs (KO). Data, mean ± SEM. Statistical significance, two-tailed Student's t-test.

DOI: https://doi.org/10.7554/eLife.29854.030

The following source data is available for figure 10:

**Source data 1.** This spreadsheet contains the data for all quantitative evaluations shown in the different panels of *Figure 10* (i.e. of *Figure 10c,d,g-l and n-t*).

DOI: https://doi.org/10.7554/eLife.29854.031

dependent signs of muscle damage. H&E stainings of *musculus gastrocnemius* cryosections from *syndapin III* KO mice subjected to physical exercise showed necrotic fibers invaded by immune cells as well as inflammation secondary to degeneration (*Figure 11c–f*; asterisk in *Figure 11c*). Acidic phosphatase analyses highlighting the lysosomal activity of immune cells proved that necrotic events occur in muscles of trained *syndapin III* KO mice (*Figure 11e*; asterisks). In contrast, irrespective of whether mice were trained or not, WT muscles did not display a single case of necrosis (*Figure 11b, d,f*).

Interestingly, *syndapin III* KO muscles were furthermore marked by a training-induced increase of detached nuclei. In humans, detached nuclei are a clear pathophysiological finding (*Gazzerro et al., 2010*). Whereas WT muscles did not show any increase in the frequency of detached nuclei upon training (about 1% in murine *musculus gastrocnemius*), detached nuclei were about three times more frequent in trained *syndapin III* KO muscles (*Figure 11g,h*).

Consistent with *syndapin III* KO-induced muscles defects, fine analyses of both untrained and trained *syndapin III* KO muscles showed altered caliber spectra (*Figure 11i,j*). Untrained *syndapin III* KO muscles showed an expansion of the caliber spectrum. Especially weakly established fibers showed a notable abundance in *syndapin III* KO mice (*Figure 11i,k*).

Interestingly, this increased abundance of fibers with only small cross-sections in *syndapin III* KO mice was compensated for by hypertrophy upon training (*Figure 11i–l*). This training-induced hypertrophy of *syndapin III* KO muscles, however, obviously was accompanied with increased rates of detached nuclei (*Figure 11h*) and of inflammation and necrosis (*Figure 11e*).

Thus, *syndapin III* KO-induced impairment of caveolar invagination was associated with observations reminiscent of the human skeletal muscle diseases linked to *CAV3* mutation.

## Discussion

Caveolae can be found in almost all body cells; yet, their functions still remain matter of debate. For decades, they were considered as endocytic structures. More recent data suggested that caveolae may represent signaling hubs or mechanosensing membrane reservoirs (*Sinha et al., 2011*). Model systems with caveolin-deficiency are unable to distinguish between these possibilities, as they lack both invaginated caveolae and caveolins (*Hansen and Nichols, 2010*; *Nassoy and Lamaze, 2012*; *Parton and del Pozo, 2013*; *Shvets et al., 2014*; *Cheng and Nichols, 2016*). Also KO of the caveolin-associated component CAVIN-1 merely mirrored cav1-deficiencies by leading to reduced caveolin levels (*Hill et al., 2008*; *Liu and Pilch, 2008*; *Liu et al., 2008*; *Bastiani et al., 2009*).

Dissecting the functions of caveolar invaginations from those of caveolins in vivo would require a situation, in which caveolins are maintained at the plasma membrane and nevertheless caveolar invaginations are severely impaired. This demand seemed impossible to meet, as overexpression experiments in bacteria suggested that caveolins are not only crucial but also sufficient for caveolae formation (*Walser et al., 2012*). Our quantitative analyses of *syndapin III* KO cardiomyocytes, heart and skeletal muscles show that *syndapin III* KO does not affect the presence of cav3 at the plasma

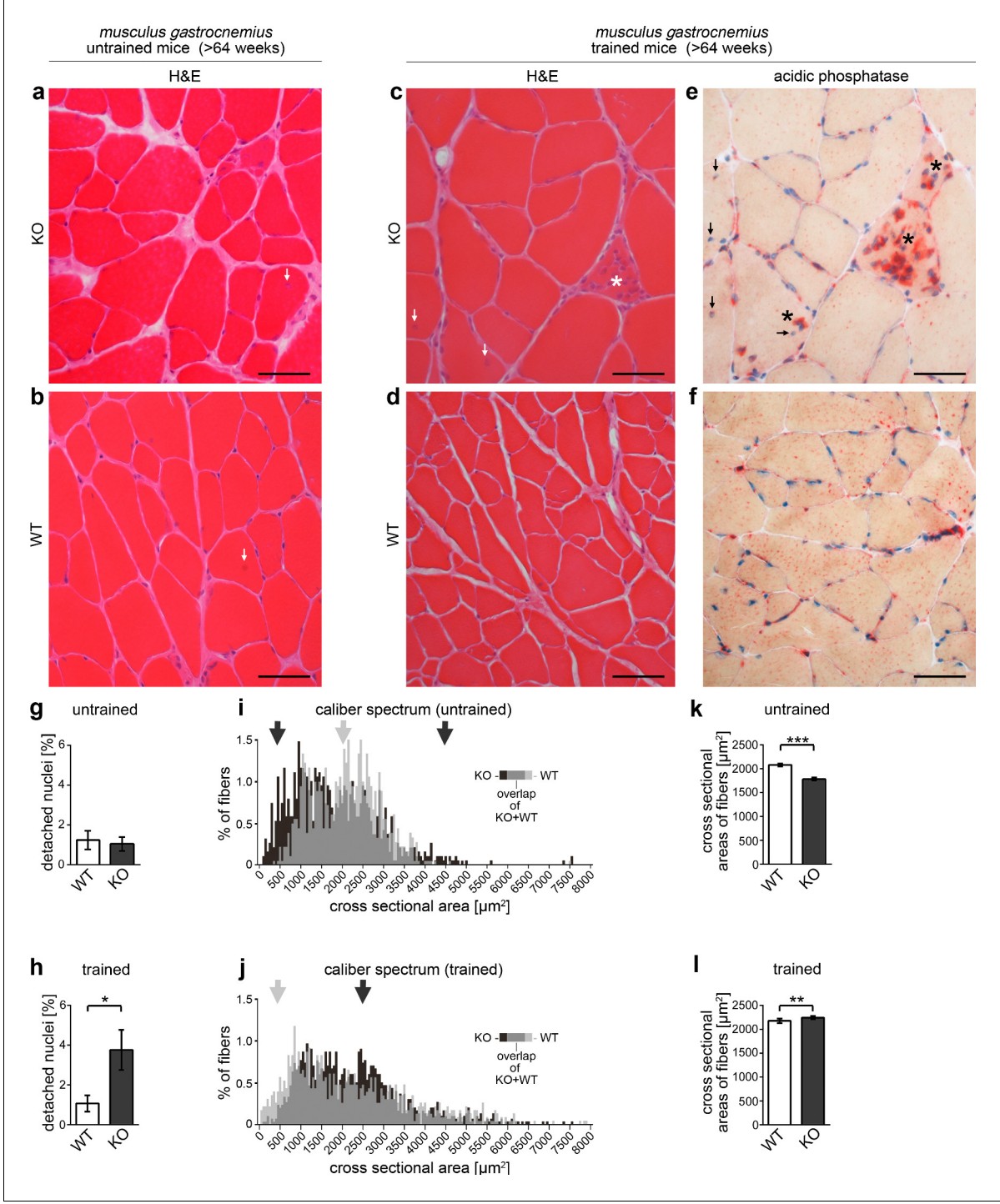

**Figure 11.** *Syndapin III* KO leads to skeletal muscle phenotypes reminiscent of clinical symptoms found in patients with myopathies associated with *CAV3* mutation. (a–f) Histological examinations of cryosections of *musculus gastrocnemius* from >64 weeks old WT and *syndapin III* KO mice (three animals each). Whereas no clear signs of cellular damage or disintegration were observed by H&E staining under untrained conditions (a,b), *syndapin III* KO mice displayed a higher frequency of detached nuclei upon training (c), arrows). (e,f) Acidic phosphatase stainings clearly demonstrate necrotic events (asterisks in c) and e). Bars, 50 µm. (g,h) Quantitative evaluations of the frequency of detached nuclei. (i,j) Percental distribution of fiber cross sectional areas (in 50 µm² intervals) in WT and *syndapin III* KO muscles. Arrows highlight areas of the caliber spectra with striking differences (black, more frequent in *syndapin III* KO; grey, more frequent in WT). (k,l) Quantitation of the mean cross sectional areas of muscle fibers in trained and untrained mice of both genotypes. N-numbers (g–l), untrained WT, 849; untrained KO, 952; trained WT, 949; trained KO, 1244 fibers. Data, mean (i,j); error bars omitted for clarity) and mean ± SEM (g,h,k,l), respectively. Statistical significance, 2-tailed Student's t-test.

DOI: https://doi.org/10.7554/eLife.29854.032

membrane but specifically impaired caveolar invagination to an extent resembling the reductions of caveolae found in human patients with *CAV3* mutations (*Gazzerro et al., 2010*; *Alias et al., 2004*; *Timmel et al., 2015*; *Minetti et al., 1998*; *Minetti et al., 2002*). These observations finally made it possible to dissect the functions of caveolae as cell biological structure from those of caveolin proteins at the plasma membrane and to thereby address the physiological relevance of caveolar invaginations at the whole animal level. Although it is a general limitation of all gene KO studies that phenotypes, such as those observed for *cav3* or *syndapin III* KO at caveolae, formally never prove a direct involvement of the lacking protein at these sites, several lines of experimental data support the conclusion that the dramatic caveolar phenotypes we observed in *syndapin III* KO heart and muscle cells reflect a direct involvement of syndapin III in the underlying, required membrane shaping process. First, syndapin III colocalized with cav3-enriched membrane domains at the light microscopical level and at cav3-marked caveolae at the ultra-structural level, that is showed the localization required for caveolar function. Second, syndapin III interacts with cav3 in coprecipitation experiments and the syndapin III F-BAR domain enriched together with cav3 in distinct plasma membrane areas. Efficient subcellular recruitment of interaction partners to membranes has also been observed for other syndapin family members (*Braun et al., 2005*; *Kessels and Qualmann, 2002*; *Schwintzer et al., 2011*; *Schneider et al., 2014*; *Hou et al., 2015*). Third, syndapins associate with phosphatidylserine using their F-BAR-domains (*Qualmann et al., 2011*; *Itoh et al., 2005*; *Dharmalingam et al., 2009*) and caveolar microdomains are enriched for phosphatidylserine (*Fairn et al., 2011*). This suggests that syndapin III-mediated membrane associations and nanocluster formation (*Schneider et al., 2014*) may help to establish membrane domains promoting cav3 assembly. In support of this, syndapin III floated with cav3-enriched DRMs and syndapin III deficiency changed the biophysical properties of cav3-associated DRMs. Fourth, an active role of syndapin III in this process is suggested by syndapin III's ability to induce tubule formation. The induced membrane tubules were about 80 nm in diameter and thereby fitted with membrane curvatures found at caveolae. Fifth, syndapin III localized specifically to necks of cav3-coated caveolar invaginations – sites, which can be expected to represent zones of curvature modulation during invagination. Finally, caveolar necks have a complex topology with negative membrane curvatures being turned into positive ones. Syndapins have a unique, curved but also tilde-like shape and may thus especially prefer and/ or promote the complex membrane topologies found at caveolar necks (*Qualmann et al., 2011*). Syndapin III thus seems to use its membrane-shaping properties to promote caveolar invagination and caveolar organization. It is conceivable that a key step in this process will be the initiation of membrane curvature in at first minimal membrane areas. This would explain how few syndapin III molecules at the neck of caveolae would suffice for triggering invagination and why not all caveolae at all times were syndapin III-positive. It is likely that this function critically required for caveolar invagination can then be propagated further by the self-assembly of cav3 (and other coat components) into bent coats.

Whether CAVIN-1 plays some role in such propagation or stabilization of caveolar invaginations or whether it simply exerts its critical role by stabilizing caveolin proteins can neither be answered by the conducted CAVIN-1 loss-of-function studies (*Hill et al., 2008*; *Liu and Pilch, 2008*; *Liu et al., 2008*) nor by our phenotypical analyses of *syndapin III* KO. Yet, two somewhat unexpected conclusions can be drawn firmly. First, even a combination of the two coat proteins cav3 and CAVIN-1 – both not reduced, that is still available, at the plasma membrane in *syndapin III* KO muscle cells - is not sufficient for efficient caveolar invagination. Second, in contrast to previous immunofluorescence studies with overexpressed CAVIN-1-mCherry and cav1-EGFP in HeLa cells (*Sinha et al., 2011*), our quantitative ultra-high-resolution studies show that CAVIN-1 is not dissociating from the plasma membrane of muscle cells when caveolae are flat but CAVIN-1 remained and its levels even increased at the plasma membrane. Our data thereby seem to be in line with a previous report showing only slightly increased cav1/CAVIN-1 ratios together with strongly increased cav1 values, which thus can be interpreted as increased CAVIN-1 levels in the TIRF zone of fixed, syndapin II RNAi-transfected HeLa cells (*Hansen et al., 2011*). Furthermore, CAVIN-1 could still be detected in cav3-enriched nanodomains, which also remained present at the predominantly flat plasma membranes found in syndapin III KO cells.

It remains to be shown whether the apparent discrepancy of our quantitative data showing that both cav3 and CAVIN-1 are still available at the plasma membrane, when membrane topologies are flat, with a part of the literature on cav1-based caveolae suggesting a dissociation or dispersion of

these coat components from their membrane nanodomains is caused by the improved resolution in our study, by the fact that we studied the endogenous proteins and/or is a specialty of cav3-caveolae and muscle cells.

It will furthermore be interesting to evaluate in detail whether similar to cav3 also cav1/cav2 in non-muscle cells would critically rely on membrane-shaping factors, too. Reports of recent years suggest that these components could for example be syndapin II and/or its binding partners, the EHD proteins (*Hansen et al., 2011*; *Morén et al., 2012*; *Yeow et al., 2017*).

Caveolae were hypothesized to act as mechanosensitive signaling hubs (*Shvets et al., 2014*). A significant portion of the plethora of physiological defects in *caveolin*-deficient mice may be mediated by MAPK hyperactivation (*Woodman et al., 2002*). We neither observe such hyperactivation nor any changes in dystrophin and β-dystroglycan expression levels or distribution upon impairment of caveolar invagination by *syndapin III* KO. We, furthermore, did not observe cardiac myocyte hypertrophy and left ventricular wall thickening in *syndapin III* KO mice. Since *syndapin III* KO did not mirror these *cav3* KO phenotypes, they either require a complete loss of all caveolae or they reflect a lack of cav3 protein as membrane-associated hub rather than dysfunctions of caveolae as invaginated cell biological structure. It thus is attractive to consider the idea that not all phenotypes and clinical symptoms observed upon cav3 deficiency may be due to currently untreatable caveolar dysfunctions, but to quite some extent may represent derailed, that is presumable treatable, signaling processes. Also the fact that even LGMD-1C patients with the severest cases of *CAV3* missense mutations known (63TFT65del and P104L) showing full clinical symptoms do not exhibit a complete impairment of caveolae formation (*Minetti et al., 1998*; *Minetti et al., 2002*) clearly argues in favor of the view that a complete loss of caveolae is not a prerequisite for the development of caveolinopathy symptoms. Instead, it seems that a severe reduction is sufficient for mirroring the caveolar defects in caveolinopathies. *Syndapin III* KO led to a severe impairment of caveolar invagination but did not alter cav3 levels at the plasma membrane of isolated cardiomyocytes, heart and skeletal muscle and thereby represents a valuable model for dissection of caveolae and caveolin functions.

Our work shows that cav3-coated caveolae serve as membrane reservoirs of muscle cells and that syndapin III plays a crucial role in forming a membrane buffer reservoir represented by invaginated caveolae. A lack of these membrane reservoirs within the plasma membrane can be expected to render cells vulnerable to all conditions associated with mechanic stress. Indeed hypoosmotic swelling caused *syndapin III* KO cardiomyocytes to rupture, whereas WT cells responded by caveolar flattening and thereby counteracted the membrane tensions induced.

Muscles are faced with abrupt changes of membrane tension and *CAV3* mutations cause muscle diseases, such as LGMD-1C and RMD (*Gazzerro et al., 2010*). Interestingly, caveolae morphology and density at the plasma membrane of rat myofibers were found to be unchanged upon different modes of contractile activity as well as upon myofiber stretching within normal myofiber limits but were greatly reduced by the more severe mechanical forces occurring during tearing of myofibers (*Poulos et al., 1986*; *Dulhunty and Franzini-Armstrong, 1975*). The muscle fibers of untrained *syndapin III* KO mice displayed an extended caliber spectrum being a sign of impaired muscle integrity. Yet, *syndapin III* KO-induced impairment of caveolar invagination did not have dramatic consequences for muscular integrity as long as mice were just housed in cages. When mice were subjected to physical exercise, however, the severe reduction of deeply invaginated caveolar structures upon *syndapin III* KO was associated with observations of hypertrophy, centronuclear myopathy and necrosis, that is were reminiscent of the clinical symptoms of human muscle diseases linked to *CAV3* mutation (*Gazzerro et al., 2010*).

Taken together, it seems that muscular caveolar invaginations are largely dispensable for life. This is in line with the viability of *cav3* KO mice (*Hagiwara et al., 2000*; *Galbiati et al., 2001*; *Woodman et al., 2002*). Yet, our studies unveil that muscular caveolae help to preserve muscle cells from the abrupt tear forces. Our data thereby shed light on the exact cellular functions impaired in caveolinopathies and strongly suggest that, when caveolae as membrane tension compensators are impaired, physical exercise-inducible hypertrophy is counterproductive to compensate the weakness of muscle fibers.

# Materials and methods

**Key resources table**

| Reagent type (species) or resource | Designation | Source or reference | Identifiers | Additional information |
|---|---|---|---|---|
| Gene (mouse and rat) | PACSIN3/syndapin III | this paper | | |
| Strain, strain background (*Mus musculus*) | C57BL/6J | Jackson Labs (Bar Harbor, Maine) | IMSR_JAX: 000664 | |
| Genetic reagent (*Mus musculus*) | 129/SvJ mouse genomic λ library | Stratagene (San Diego, California) | | |
| Cell line (*Mus musculus*) | NIH3T3 | Cell Lines Services GmbH (Eppelheim, Germany) | CVCL_0594 | |
| Cell line (human) | HEK293 | Cell Lines Services GmbH | CVCL0045 | |
| Transfected construct (rat) | GFP-syndapin III F-BAR (aa 1–336; extended F-BAR) | this paper | | |
| Transfected construct (mouse) | Syndapin III aa1-70 peptide (plus five unrelated aa encoded by exon5/6 deletion-induced frame shift and multiple stop codons) | this paper | | |
| antibody | guinea pig anti-syndapin I, rabbit anti-syndapin II, rabbit anti-syndapin III, anti-GST | *Koch et al. (2011)*, EMBO J 30:4955–4969., *Qualmann et al. (1999)*, Mol Biol Cell 10:501–513. *Koch et al. (2012)*, Histochem Cell Biol 138: 215-230., *Qualmann and Kelly (2000)*, J Cell Biol 148:1047–1062 | | anti-syndapin I, II, III 1:1000 (western blot); 1:50 (FRIL) |
| Antibody | goat anti-GAPDH (polyclonal) | Santa Cruz (Dallas, Texas) | sc48167 AB_1563046 | 1:1000 |
| Antibody | mouse anti-cav3 (monoclonal ) | Santa Cruz | sc-5310, AB_626814 | 1:500 (western blot), 1:50 (IHC), 1:50 (FRIL) |
| Antibody | goat anti-cav3 (polyclonal) | Santa Cruz | sc-7665 AB_637945 | 1:500 (western blot), 1:50 (IHC) |
| Antibody | mouse anti-cav1 (monoclonal) | Santa Cruz | sc-53564, AB_628859 | 1:500 (western blot), 1:50 (IHC), 1:50 (FRIL) |
| Antibody | rabbit anti-cav1 (polyclonal) | Santa Cruz | sc-894 AB_2072042 | 1:1000 (western blot), 1:200 (IHC), 1:50 (FRIL) |
| Antibody | rabbit anti-IRTK (polyclonal) | Santa Cruz | sc-710, AB_631106 | 1:1000 (western blot) |
| Antibody | mouse anti-CAVIN-1 (monoclonal) | BD Bioscience (Franklin Lakes, New Jersey) | 611258, AB_398788 | 1:50 (FRIL) |
| Antibody | mouse anti-PDI (monoclonal) | Stressgene (Farmingdale, New York) | ADI-SPA-891, AB_10615355 | 1:1000 (western blot) |
| Antibody | mouse anti-β-actin (monoclonal) | Sigma | A5441, AB_476744 | 1:5000 (western blot) |
| Antibody | mouse anti-β-tubulin (monoclonal) | Sigma | T4026, AB_477577 | 1:1500 (western blot) |
| Antibody | rabbit anti-CAVIN-1 (polyclonal) | Proteintech (Rosemont, Illinois) | 18892–1-AP, AB_10596795 | 1:1000 (western blot), 1:100 (IHC), 1:50 (FRIL) |
| Antibody | rabbit anti-CAVIN-3 (polyclonal) | Proteintech | 16250–1-AP AB_2171894 | 1:1000 (western blot) |
| Antibody | goat anti-CAVIN-2 (polyclonal) | R & D Systems (Minneapolis, Minnesota) | AF5759 AB_2269901 | 1:200 (western blot) |
| Antibody | rabbit anti-CAVIN-4 (polyclonal) | Sigma | HPA020987 AB_1853080 | 1:400 (western blot) |

*Continued on next page*

*Continued*

| Reagent type (species) or resource | Designation | Source or reference | Identifiers | Additional information |
|---|---|---|---|---|
| Antibody | mouse anti-β-dystroglycan (monoclonal) | Leica Biosystems (Wetzlar, Germany) | NCL-b-DG, AB_442043 | 1:1000 (western blot); 1:200 (IHC) |
| Antibody | mouse anti-dystrophin (monoclonal) | Leica Biosystems | NCL-DYS1, AB_442080 | 1:200 (western blot); 1:50 (IHC) |
| Antibody | mouse anti-GM130 (monoclonal) | BD Biosciences | 610822 AB_398141 | 1:1000 (western blot) |
| Antibody | mouse anti-β-catenin (monoclonal) | BD Biosciences | 610153, AB_397554 | 1:1000 (western blot) |
| Antibody | mouse anti-ERK1/2 (monoclonal) | Cell Signalling (Danvers, Massachusetts) | #9107 AB_2235073 | 1:5000 (western blot) |
| Antibody | mouse anti-pERK1/2 (monoclonal) | Cell Signalling | #9106, AB_331768 | 1:1000 (western blot) |
| Antibody | mouse anti-GFP (monoclonal) | Clontech (Mountain View, California) | 632380, AB_10013427 | 1:8000 (western blot) |
| Antibody | rabbit anti-Cherry (polyclonal) | Abcam (Cambridge, UK) | ab167453 AB_2571870 | 1:1000 (western blot) |
| Antibody | Alexa Fluor555-labeled wheat germ agglutinin (WGA) | Molecular Probes (Eugene, Oregon) | W32464 | 1:2000 (IF) |
| Antibody | Alexa Fluor488-labeled wheat germ agglutinin (WGA) | Molecular Probes | W11261 | 1:2000 (IF) |
| Antibody | Alexa Fluor488-labeled goat anti-guinea pig | Molecular Probes | AB_142018 | 1:1000 (IF) |
| Antibody | Alexa Fluor568-labeled goat anti-guinea pig | Molecular Probes | AB_141954 | 1:1000 (IF) |
| Antibody | Alexa Fluor488-labeled donkey anti-mouse | Molecular Probes | AB_141607 | 1:1000 (IF) |
| Antibody | Alexa Fluor568-labeled donkey anti-mouse | Molecular Probes | AB_2534013 | 1:1000 (IF) |
| Antibody | Alexa Fluor647-labeled goat anti-mouse | Molecular Probes | AB_141725 | 1:1000 (IF) |
| Antibody | Alexa Fluor488-labeled donkey anti-rabbit | Molecular Probes | AB_141708 | 1:1000 (IF) |
| Antibody | Alexa Fluor568-labeled goat anti-rabbit antibodies | Molecular Probes | AB_143011 | 1:1000 (IF) |
| Antibody | Alexa Fluor647-labeled goat anti-rabbit antibodies | Molecular Probes | AB_141775 | 1:1000 (IF) |
| Antibody | Alexa Fluor680-labeled donkey-anti-goat | Molecular Probes | AB_141494 | 1:10000 (western blot) |
| Antibody | Alexa Fluor680-labeled goat-anti-rabbit | Molecular Probes | AB_2535758 | 1:10000 (western blot) |
| Antibody | Alexa Fluor680-labeled goat-anti-mouse | Molecular Probes | AB_1965956 | 1:10000 (western blot) |
| Antibody | goat anti-rabbit-peroxidase | Dianova (Hamburg, Germany) | AB_2337945 | 1:10000 (western blot) |
| Antibody | goat anti-guinea pig-peroxidase | Dianova | AB_2337405 | 1:10000 (western blot) |
| Antibody | goat anti-mouse-peroxidase | Dianova | AB_2338523 | 1:10000 (western blot) |
| Antibody | DyLight800-conjugated goat anti-rabbit | Pierce/Thermo (Waltham, Massachusetts) | AB_2556775 | 1:10000 (western blot) |
| Antibody | DyLight800-conjugated goat anti-mouse | Pierce | AB_2556774 | 1:10000 (western blot) |
| Antibody | IRDy 800CW-conjugated donkey anti-goat | BioTrend (Köln, Germany) | AB_220102 | 1:10000 (western blot) |

*Continued on next page*

*Continued*

| Reagent type (species) or resource | Designation | Source or reference | Identifiers | Additional information |
|---|---|---|---|---|
| Antibody | anti-guinea pig antibodies coupled to IRDye680 | LI-COR Bioscience (Lincoln, Nebraska) | AB_10956079 | 1:10000 (western blot) |
| Antibody | anti-guinea pig antibodies coupled to IRDye800 | LI-COR Bioscience | AB_1850024 | 1:10000 (western blot) |
| Antibody | gold-labeled goat anti-rabbit (10 nm) | British Biocell International (Cardiff, UK) | AB_1769130 | 1:50 (FRIL) |
| Antibody | gold-labeled goat anti-rabbit (15 nm) | British Biocell International | AB_1769134 | 1:50 (FRIL) |
| Antibody | gold-labeled goat anti-mouse (10 nm) | British Biocell International | AB_1769156 | 1:50 (FRIL) |
| Antibody | gold-labeled goat anti-mouse (15 nm) | British Biocell International | AB_2715551 | 1:50 (FRIL) |
| Recombinant DNA reagent | rat syndapin III full length (aa 1–424) in pGEX-5X1 (plasmid) | *Braun et al. (2005)* Mol Biol Cell, 16:3642–3658. | | |
| Recombinant DNA reagent | rat syndapin III F-BAR (aa 1–336; extended F-BAR) in pGEX-5X1 (plasmid) | this paper | | |
| Recombinant DNA reagent | rat syndapin III ΔF-BAR (aa 336–424) in pGEX-5X1 (plasmid) | this paper | | |
| Recombinant DNA reagent | rat syndpin III SH3 (aa 366–424) in pGEX-5X1 (plasmid) | this paper | | |
| Recombinant DNA reagent | rat syndapin III ΔSH3 (aa 1–365) in pGEX-5X1 (plasmid) | this paper | | |
| Recombinant DNA reagent | GFP-rat syndapin III F-BAR (aa 1–336; extended F-BAR) in pEGFP-C2 (plasmid) | this paper | | |
| Recombinant DNA reagent | mouse syndapin III aa1-70 peptide (plus five unrelated aa encoded by exon5/6 deletion-induced frame shift and multiple stop codons) plasmids including pGEM-T, mCherry-pCMV-Tag2b, pEGFP-C2 and pGEX-5X-1 | this paper | | |
| Peptide, recombinant protein | rat GST-syndapin III full length (aa 1–424) and rat syndapin III full-length (untagged) | *Braun et al. (2005)* Mol Biol Cell 16:3642–3658 (for plasmid and (uncut) GST-fusion protein) | | Untagged syndapin III was obtained from GST-syndapin III upon cleavage of the GST tag with 6 U precission protease/mg protein and overnight dialysis against 20 mM HEPES, 150 mM NaCl, 2 mM EDTA und 2.5 mM DTT pH 7.4 at 4°C. |
| Peptide, recombinant protein | rat GST-syndapin III F-BAR (GST+Sdp III aa 1–336; extended F-BAR) | this paper | | |
| Peptide, recombinant protein | rat GST-syndapin III ΔF-BAR (GST+Sdp III aa 336–424) | this paper | | |

*Continued on next page*

*Continued*

| Reagent type (species) or resource | Designation | Source or reference | Identifiers | Additional information |
|---|---|---|---|---|
| Peptide, recombinant protein | rat GST-syndapin III SH3 (GST+SdpIII aa 366–424) | this paper | | |
| Peptide, recombinant protein | rat GST-syndapin III ΔSH3 (GST+Sdp III aa 1–365) | this paper | | |
| Commercial assay or kit | NucleoSpin Plasmid | Macherey-Nagel (Düren, Germany) | 740.588.250 | |
| Commercial assay or kit | NucleoBond Xtra Midi | Macherey-Nagel | 740.410.100 | |
| Commercial assay or kit | Rediprime II Random Prime Labelling System | GE Healthcare (Chicago, Illinois) | #RPN1633 | |
| Chemical compound, drug | α P$^{32}$ dCTP | GE Healthcare (Chicago, Illinois) | | |
| Software, algorithm | AxioVision 4.8.2 | Zeiss (Oberkochen, Germany) | SCR_002677 | |
| Software, algorithm | ZEN 2012 | Zeiss | SCR_013672 | |
| Software, algorithm | Prism6 | GraphPad (La Jolla, California) | SCR_002798 | |
| Software, algorithm | imageJ | other | SCR_003070 | open source software |
| Software, algorithm | IMOD package | *Kremer et al. (1996)* Journal of Structural Biology, 116, 71–76 | SCR_003297 | open source software |
| Software, algorithm | IMARIS 8.4 | Bitplane (Zürich, Switzerland) | SCR_007370 | |
| Other | DAPI stain | Molecular Probes | | (1:10000) |

## Generation of *syndapin III KO* mice

The targeting vector was constructed using a clone isolated from a 129/SvJ mouse genomic λ library (Stratagene). An BamHI/blunt/HindIII fragment of approximately 2.6 kbp of this clone including exons 1–2 of the *syndapin III* gene was combined with a PCR-generated, approximately 4 kbp large HindIII/SnaBI fragment including exons 3–11 and cloned into the pKO-DTA plasmid (Lexicon Genetics) containing a phosphoglycerate kinase (pgk) promoter-driven diphtheria toxin A cassette.

A pgk promoter-driven neomycin resistance cassette flanked by frt sites and a loxP site was inserted into the DraIII site of intron 6. A second loxP site together with an additional HindIII site was inserted into the EcoRI site in intron 4. The EcoRI site was destroyed upon insertion.

R1 mouse embryonic stem (ES) cells were electroporated with NotI-linearized targeting vector. Neomycin-resistant ES cell clones were analyzed by Southern blotting. A P$^{32}$-labeled 425-bp probe (NC_000068.6 *Mus musculus* chromosome 2, nt4262-4686) was used to analyze EcoRI-digested DNA isolated from 288 neomycin-resistant ES cell clones (WT, 4,500 bp band, nt 3928–8419; transgene, approx. 5,700 bp band, nt 3928–9532 + 130 bp piece of neo cassette due to a EcoRI site in the neo cassette).

Two correctly targeted ES cell clones (G10, D6) were injected into C57BL/6J blastocysts to generate chimeras. *Syndapin III KO* mice were obtained via mating with mice ubiquitously expressing modified Cre recombinase under the transcriptional control of the human cytomegalovirus minimal promoter P$_{Bi-2}$ (*Schwenk et al., 1995*) to remove exons 5 and 6 together with the selection cassette.

For genotyping, DNA of mouse tail biopsies was extracted using 10 mM Tris/HCl pH 8.0, 100 mM NaCl, 0.8 mg/ml proteinase K (overnight, 55°C). After inactivation of the reaction (10 min, 95°C), a high-speed supernatant (5 min, 20,000 x g) was analyzed by PCR (Fwd1, 5′-cacgttagcaatggcagtgt-3′; Rev1, 5′-ggcgggtttgctaacttaaa-3′ and Rev2, 5′-tgttggaggcagaatgtcaa-3′). The primer pair Fwd1/Rev1 amplified a 206 bp WT allele, and the primer pair Fwd1/Rev2 a 164 bp KO allele. RNA isolation and reverse transcription PCR was done as described before (*Haag et al., 2012*). Two micrograms of DNase-treated total RNA from heart and skeletal muscle were reverse transcribed using oligo(dT)-primers and RevertAid H Minus Reverse Transcriptase. Controls omitting reverse transcriptase (- Rev.Tr.) were run in parallel. *Syndapin III* expression was analyzed by PCR using the following primers: *syndapin III* Fwd, 5′-

atggctccagaggaggacgcc-3'; Rev, 5'-cccccccagcactggccg-3' and *GAPDH* Fwd, 5'-attgacctcaacta-catggtctaca-3'; Rev, 5'-ccagtagactccacgacatactc-3'.

All animal procedures were performed in strict compliance with the EU directives 86/609/EWG and 2007/526/EG guidelines for animal experiments and were approved by the local government (permission numbers: 02-011/10 for generation and establishment of *syndapin III* KO mice; 02-057/14 (*18.09.2014*)/*Änderungsbescheid I (19.02.2015)* for motor training; Thüringer Landesamt für Verbraucherschutz, Bad Langensalza; Germany).

Animals were housed under 14 hr light/10 hr dark conditions with ad libitum access to food and water. *Syndapin III* KO (-/-) mice were backcrossed on C57BL/6J.

Trained mice (age matched WT and *syndapin III* KO male mice with a C57BL/6J background of >93.7%) received 10 running sessions at a speed of 20 cm/s (1 hr per mouse and per week) prior to histological examination of muscle integrity.

## Antibodies

Anti-syndapin III, anti-syndapin II, anti-syndapin I and anti-GST antibodies and their affinity purification were described previously (*Koch et al., 2011*; *Koch et al., 2012*; *Qualmann and Kelly, 2000*; *Qualmann et al., 1999*). Polyclonal goat anti-GAPDH antibodies (sc48167), monoclonal mouse and polyclonal goat anti-cav3 (sc-5310; sc-7665), monoclonal mouse and polyclonal rabbit anti-cav1 (sc-53564; sc-894) as well as polyclonal rabbit anti-IRTK antibodies (sc-710) were from SantaCruz. Anti-CAVIN-1 antibodies (611258) were from BD Bioscience. Monoclonal anti-PDI antibodies (ADI-SPA-891) were from Stressgene. Monoclonal mouse anti-β-actin (A5441) and anti-β-tubulin (T4026) antibodies were from Sigma. Polyclonal rabbit anti-CAVIN-1 and 3 antibodies (18892–1-AP; 16250–1-AP) were from Proteintech, polyclonal goat anti-CAVIN-2 antibodies (AF5759) were from R&D Systems and polyclonal rabbit anti-CAVIN-4 antibodies (HPA020987) were from Sigma. Monoclonal mouse anti-β-dystroglycan and anti-dystrophin antibodies were from Leica (NCL-b-DG; NCL-DYS1). Monoclonal anti-GM130 and anti-β-catenin antibodies were from BD Biosciences (610822; 610153). Monoclonal mouse anti-ERK1/2 and anti-pERK1/2 (#9107; #9106) were from Cell Signaling. Monoclonal mouse anti-GFP (632380) and polyclonal anti-mCherry antibodies (ab167453) were from Clontech and Abcam, respectively.

Alexa Fluor®555- and Alexa Fluor 488-labeled wheat germ agglutinin (WGA) was purchased from Molecular Probes. Fluorescently labeled secondary antibody conjugates used for this study included, Alexa Fluor®488- and 568-labeled goat anti-guinea pig antibodies, Alexa Fluor®488 and 568-labeled donkey anti-mouse as well as Alexa Fluor®647-labeled goat anti-mouse antibodies, Alexa Fluor®488-labeled donkey anti-rabbit, Alexa Fluor®568 and 647-labeled goat anti-rabbit antibodies and Alexa Fluor680-labeled donkey-anti-goat, goat anti-rabbit and goat anti-mouse antibodies (Molecular Probes); goat anti-rabbit, anti-guinea pig and anti-mouse-peroxidase antibodies (Dianova); DyLight800-conjugated goat anti-rabbit and anti-mouse antibodies (Pierce), IRDye800CW-conjugated donkey anti-goat antibodies (BioTrend) anti-guinea pig antibodies coupled to IRDye680 and IRDye800, respectively, (LI-COR Bioscience).

Gold-labeled goat anti-rabbit (10 nm and 15 nm) and goat anti-mouse (10 nm and 15 nm) secondary antibodies were from British Biocell International.

## Plasmids and recombinant fusion proteins

Syndapin III full length (aa 1–424) was cloned from a rat cDNA library and inserted into pGEX-5X1 as described previously (*Braun et al., 2005*). Syndapin III F-BAR (aa 1–336; extended F-BAR), ΔF-BAR (aa 336–424), SH3 (aa 366–424) and ΔSH3 (aa 1–365) were generated by PCR with appropriate primers and GST-syndapin III as template and were cloned into pGEX vectors. All PCR-generated constructs were sequenced to verify that their sequences were correct.

GST fusion proteins were prepared as described previously (*Qualmann et al., 1999*). Untagged syndapin III was obtained from GST-syndapin III upon cleavage of the GST tag with 6 U precission protease/mg protein as described (*Zobel et al., 2015*) and overnight dialysis against 20 mM HEPES, 150 mM NaCl, 2 mM EDTA und 2.5 mM DTT pH 7.4 at 4°C.

GFP-syndapin III F-BAR (aa 1–336; extended F-BAR) was generated by subcloning into pEGFP-C2.

Plasma-membrane-targeted fluorescent proteins were described previously (*Schneider et al., 2014*).

Syndapin III[1-70] peptide (plus five unrelated aa encoded by exon5/6 deletion-induced frame shift and multiple stop codons) was generated by PCR and cloning into pGEM-T (Promega). The construct was subcloned (EcoRI/SalI) into mCherry-pCMV-Tag2b, pEGFP-C2 and pGEX-5X1, respectively.

## Morphological analysis of cardiac myocyte cross-sections and ventricular wall thickness

To analyze cardiac myocyte cross-sections, 20 weeks old WT and *syndapin III* KO mice were killed by cervical dislocation. Beginning at 1000 µm from the apex, hearts were cut in 8 µm thick slices using a cryostat (CM3050S Leica Biosystems).

Cardiomyocytes lining the left ventricle from at least two sections per animal were stained with WGA and cardiac myocyte cross-sections were determined (Fluar 40x/1.30 Oil M27 objective; Zeiss Apotome2/Axio Observer.Z1; AxioCam MR Rev3 camera). The analysis was done using ImageJ software (ImageJ).

Left ventricular wall thickness was determined using cardiac cryosections from 20 weeks old mice stained with H&E that were imaged by using a EC Plan-Neofluar 5x/0.16 objective (microscope and camera as above). Pictures were taken as tiles with 20% overlap. Based on a stitched image (ZEN 2012 software (Zeiss)), the wall thickness was calculated as half the difference of the outer and inner diameter of the left ventricle using the ZEN 2012 software (Zeiss).

## Immunofluorescence analyses of heart and skeletal muscle sections

Heart and *musculus quadriceps* and *gastrocnemius* samples, respectively, were from 12 to 15 weeks old WT and *syndapin III* KO mice. After processing for cryosectioning 20 µm thick transversal and 40 µm thick longitudinal sections were blocked with 5% goat serum in 0.1 M phosphate buffer (pH 7.4) and 0.25% Triton X-100 for 1 hr and incubated with primary antibodies in the above buffer at 4°C for 12 hr. Secondary antibodies were applied for 2 hr. Subsequent to DAPI staining, sections were embedded in Fluoromount-G.

Confocal images of tissue sections were recorded digitally with a Leica TCS SP5 confocal microscope operated under identical settings.

Quantitative analyses of anti-cav3 intensities were done with ImageJ. Rectangular ROIs of about 5 × 20 µm were placed at the plasma membranes of two adjacent fibers and ROIs for intracellular measurements (complete intracellular area of a fiber) were drawn with about 5 µm distance from plasma membrane at confocal sections of transversally sectioned immunostained muscle fibers. Determinations of anti-cav3 intensities at membrane-associated puncta were done using longitudinal sections. Quantitative analyses of cav3 and syndapin III were done using the ImageJ plugin Coloc2 (version 3.0.0). ROIs were predefined in an unbiased manner in the anti-cav3 channel for all measurements and analyzed for syndapin III immunosignals subsequently.

3D surface rendering of anti-cav3 immunolabeling was done using IMARIS 8.4 (Bitplane). ROIs (14 in total, 2 per image) were defined in an unbiased manner using the anti-cav3 channel in confocal image stacks of longitudinal muscle cryosections. Cav3 spots were generated with a defined diameter of 0.5 µm. The generated spheres were then filtered for syndapin III presence or absence (expressed as percent of all cav3 spots (averaged per image) for quantitative analyses) and color-coded (for visual presentation).

## Muscle pathology examinations

Mice were killed by cervical dislocation. 7 × 7 mm biopsies from *musculus gastrocnemius* were cryo-conserved and 4% PFA-fixed/paraffin-embedded, respectively. Cryosections (10 µm) were used for H&E, trichrome, sudan black, periodic acid–Schiff, ATPase (pH 4.2, 4.6, 9.4), acidic phosphatase, succinate dehydrogenase, cytochrome oxidase, NADH-tetrazolium reductase, myoadenylate deaminase, phosphofructokinase, ATPase, iron and lactate dehydrogenase stainings. Paraffin sections (4 µm) were used for H&E, Congo Red, periodic acid–Schiff and Elastic van Gieson (EvG) stainings according to standard procedures.

Images were taken with a Zeiss Observer Z.1, a 20x/0.5 objective and AxioVision 4.8.2 software. Analyses of the percent of detached nuclei and of the caliber spectra were conducted using H&E stainings. Caliber spectra were done with ImageJ and the *select wand* and *freehand selection* tools to mark muscle fibers.

## Osmotic stress assay, transfection and immunofluorescence analyses of cardiomyocytes

Cardiomyocytes were prepared, transfected with lipofectamine and subjected to immunofluorescence analysis as described (*Lygren et al., 2007*; *Schwintzer et al., 2011*).

Cardiomyocytes from WT and *syndapin III* KO mice were subjected to iso buffer (20 mM HEPES pH 7.4, 1 mM $CaCl_2$, 1 mM $MgCl_2$, 5 mM KCl supplemented with 10.5 mM glucose, 28.8 μM BSA and 150 mM NaCl), to hypo15 buffer (iso buffer with 15 mM instead of 150 mM NaCl), and to hypo30 buffer (iso buffer with 30 mM instead of 150 mM NaCl), respectively, for 5 min. Cells incubated in iso and hypo15 buffer were analyzed by freeze-fracturing and EM (similar experiments were conducted with NIH3T3 cells). Analyses were conducted by an independent experimentator in a blinded manner.

In further assays, cardiomyocytesc from WT and *syndapin III* KO mice were subjected to iso conditions and to hypo30- and hypo15-induced mild and stronger membrane tension stress and were stained with Trypan Blue. Ruptured cells were counted systematically in a blinded study (about 100 cells/assay; 18 assays).

## Cell culture and immunofluorescence analyses of cell lines

Culturing of NIH3T3 cells and HEK293 cells (Cell Lines Services GmbH) was essentially done as described (*Kessels et al., 2001*). All cell lines are regularly tested for mycoplasma and were mycoplasma-negative. Transfections were done using TurboFect according to the instructions of the supplier (Thermo Scientific).

Fluorescence microscopy of NIH3T3 cells fixed 24 hr after transfection (4% PFA for 7 min) was done using a Zeiss AxioObserver.Z1 microscope equipped with an ApoTome2 and Plan-Apochromat 63x/1.4 and 40x/1.3 objectives and an AxioCam MRm CCD camera (Zeiss). Digital images were recorded by ZEN2012 software and image processing was done using Adobe Photoshop.

## Biochemical analyses

Tissue homogenizations and quantitative immunoblotting were essentially performed as described (*Schwintzer et al., 2011*).

Heart and skeletal muscle pieces (from *musculus quadriceps* and *gastrocnemius*) were first ground to powder under liquid $N_2$.

TritonX-100 insoluble material was obtained by centrifugation at 100000 x g for 1 hr at 4°C, washing was done by resuspension and repeated centrifugation (15 min).

For DRM preparations, homogenates were prepared in protease inhibitor-supplemented MES buffer (50 mM MES pH 6.5, 150 mM NaCl, 1 mM EDTA, 0.1 mM $MgCl_2$). After TritonX-100 addition (1% final), the homogenates were incubated with gentle shaking at 4°C for 30 min. Samples were centrifuged (1000 x g for 5 min at 4°C), and the postnuclear supernatants were adjusted to 40% sucrose by the addition of 2 ml of 80% sucrose in MES buffer, overlayed with a 5–30% linear sucrose gradient and centrifuged for 16 hr (284000 x g (40000 rpm, SW40 Ti)) and for 24 hr (174600 x g (32000 rpm, SW32 Ti)), respectively. 12 fractions à 1 ml were collected (from top). Proteins were precipitated by 75% ethanol and immunoblotted.

For immunoblotting with anti-pERK1/2 antibodies, hearts from 12 weeks old WT and *syndapin III* KO mice were homogenized in lysis buffer II (50 mM HEPES pH 7.4, 150 mM NaCl, 1 mM EDTA, 0.1 mM $MgCl_2$, protease inhibitors) with phosphatase inhibitors via sonication. 50 μg protein were immunoblotted and analyzed quantitatively in relation to anti-GAPDH and anti-ERK1/2 signals, respectively.

Fractionations addressing the membrane association of GFP, GFP-syndapin III F-BAR and of the putative GFP-syndapin III[1-70] (aa1-70 peptide of syndapin III plus five aa due to frame shift) were done with lysates of transfected HEK293 cells. After passage through a syringe with a 25G cannula in 5 mM HEPES pH 7.4, 0.32 M sucrose and 1 mM EDTA centrifugation (1000 x g, 4°C, 10 min)

resulted in fraction S1 and P1. Centrifugation of S1 (12000 x g, 4°C, 10 min) yielded fractions S2 and the crude membrane fraction P2. Cotransfected plasma membrane-targeted mCherry (mCherryF) served as internal control for a membrane-bound protein.

For coprecipitation analyses, heart homogenates in lysis buffer II were centrifuged at 100000 x g at 4°C for 30 min. Membrane-containing pellets were solubilized by 1% TritonX-100 in PBS, incubated 30 min at RT and centrifuged (20 min, 17000 x g, 4°C). Supernatants were transferred to agarose-coupled GST-fusion proteins and after 3 hr at 4°C the samples were washed, eluted with SDS sample buffer and immunoblotted.

## Protein/liposome binding assays and EM analyses of liposomes

Freeze fracturing of liposomes incubated with proteins (2 µM) was done as described (*Beetz et al., 2013*). Cryo-TEM of liposomes coated with GST-syndapin III and syndapin III and of control incubations was performed on holey carbon film-covered copper grids. Liquid ethane-frozen samples (~−180°C) were transferred into a pre-cooled cryo-transmission electron microscope operated at 120 kV (Philips CM 120) using a cryo-transfer unit (Gatan 626-DH). Images were recorded with a 1K CCD Camera (FastScan F114, TVIPS).

Freeze-fracture replica were viewed using a Zeiss EM 902A transmission electron microscope run at 80 kV (Zeiss). Images were recorded digitally using an 1 k FastScan-CCD-camera (TVIPS camera and software).

Tubule diameters were analyzed using ImageJ. For diameter distribution analyses, measured diameters were grouped in 5 nm-step categories (0–5 nm,>5 to 10 nm,>10 to 15 nm and so on).

## Cardiomyocyte fixation, sectioning and processing for EM analyses

Primary cardiomyocytes isolated from WT and *syndapin III* KO mice were fixed by adding 2.5% glutaraldehyde in 0.1 M cacodylate buffer. After 1 hr, the cells were collected by centrifugation at 600 x g and washed 3x with PBS. After post-fixation in 1% osmiumtetroxide for 1 hr, dehydration in ascending ethanol series with post-staining in uranylacetate was performed. The samples were embedded in epoxy resin (Araldite) and finally sectioned ultrathin using a LKB Ultratome III (LKB). After mounting on coated copper grids and post-staining with lead citrate for 10 min, the sections were examined in a TEM (EM 902A, Zeiss) at 80 kV. Images were recorded digitally using an 1 k FastScan-CCD-camera (TVIPS camera and software). Plasma membrane sections of about 2 µm length each were analyzed for the presence or absence of caveolar membrane profiles. Data were expressed as caveolae/length of membrane stretch. Images were processed using Adobe Photoshop software.

## Freeze-fracturing of cardiomyocytes, of NIH3T3 cells and of heart and skeletal muscle tissues

Freeze-fracturing of NIH3T3 cells and primary cardiomyocytes was essentially done as described (*Koch et al., 2012*; *Schneider et al., 2014*).

To obtain freeze-fracture replica directly from skeletal muscles and hearts, heart and *musculus gastrocnemius* were isolated and cut longitudinal in 300 µm-thick slices using a McIlwain Tissue Chopper. Sections were transferred into PBS and frozen in 20% (w/v) BSA between a copper head sandwich profile. Copper sandwiches were freeze-fractured, shadowed with carbon and platinum/carbon, and incubated in 0.2% collagenase II in DMEM at 37°C overnight. After washing, replica were incubated in 5% (w/v) SDS and 30 mM sucrose in 10 mM Tris/HCl, pH 8.4 overnight.

## Immunolabeling and TEM of freeze-fractured plasma membranes

Replica were immunolabeled and analyzed by TEM as described (*Schneider et al., 2014*). Quantitative evaluations were performed by an independent experimentator and in a blinded manner using samples from several independent freeze-fracturing experiments. Cav1, cav3, CAVIN-1 and syndapin III labeling densities were determined per 3.47 µm$^2$ image and analyzable ROI, respectively, and expressed as % of the corresponding WT and isoosmotic condition, respectively, of each assay (set at 100%). Immunolabels were only considered as localized to a caveolar invagination if localized <50 nm from the (inner) caveolar rim. In total, 1087 µm$^2$ membrane were scored for quantitative analyses and 2889 caveolin-positive invaginations were evaluated.

Cluster analyses of cav3 and CAVIN-1 immunogold signals were done using circular ROIs of 150 nm diameter around caveolae (70 nm (inner caveolar diameter) + 2×10 nm (PM curvature zone around the caveolae) + 2×30 nm (maximally possible primary/secondary antibody extension and gold particle size). Four or more immunosignals per ROI were considered as cluster.

### Electron tomography of freeze-fractured caveolae

For electron tomography, replica specimens were placed in a tilt-rotate specimen holder (Model 626; Gatan) and tomographic data sets were recorded using a Philips CM120 operated at 120 kV. Images were captured every 2° over a −70° to 56° range using a 2K CCD camera (TecCam F216, TVIPS).

The tilted views were aligned using the positions of the gold particles as fiducial points. Tomograms were computed and analyzed with the software IMOD Package (*Kremer et al., 1996*).

### Statistical analyses

Prism6 was used for testing for normal data distribution and for statistical analysis (for methods used for statistical significance calculations please see the figure legends accompanying the respective data panel). $*p<0.05$, $**p<0.01$, $***p<0.001$ throughout.

## Acknowledgements

This work was supported by grants from the *Deutsche Forschungsgemeinschaft* to CAH, MMK and BQ. We thank R Ahuja, A Büschel, K Häßler, A Hübner, A Kreusch, B Schade, M Öhler, C Scharf, K Schorr, T Laudage, F Steiniger, D Koch and D Wolf for technical help and support as well as C Hennings and J von Maltzahn for advice.

## Additional information

### Funding

| Funder | Grant reference number | Author |
|---|---|---|
| Deutsche Forschungsgemeinschaft | | Christian A Huebner |
| Deutsche Forschungsgemeinschaft | KE685/3-2 | Michael M Kessels |
| Deutsche Forschungsgemeinschaft | QU116/6-2 | Britta Qualmann |
| Deutsche Forschungsgemeinschaft | RTG1715 | Britta Qualmann |

The funders had no role in study design, data collection and interpretation, or the decision to submit the work for publication.

### Author contributions

Eric Seemann, Minxuan Sun, Sarah Krueger, Jessica Tröger, Natja Haag, Susann Schüler, Validation, Investigation, Visualization, Methodology; Wenya Hou, Investigation; Martin Westermann, Supervision, Methodology; Christian A Huebner, Bernd Romeike, Supervision; Michael M Kessels, Britta Qualmann, Conceptualization, Supervision, Funding acquisition, Validation, Visualization, Writing—original draft, Writing—review and editing

### Author ORCIDs

Michael M Kessels (iD) http://orcid.org/0000-0001-5967-0744

### Ethics

Animal experimentation: All animal procedures were performed in strict compliance with the EU directives 86/609/EWG and 2007/526/EG guidelines for animal experiments and were approved by the local government (permission numbers: 02-011/10 for generation and establishment of syndapin

III KO mice; 02-057/14 (18.09.2014)/Änderungsbescheid I (19.02.2015) for motor training; Thüringer Landesamt, Bad Langensalza; Germany).

### Decision letter and Author response
Decision letter https://doi.org/10.7554/eLife.29854.035
Author response https://doi.org/10.7554/eLife.29854.036

## Additional files

**Supplementary files**
• Transparent reporting form
DOI: https://doi.org/10.7554/eLife.29854.033

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
