## [Decision Letter]

[Editor's note: Please note that due to the subsequent change of two figures into the figure supplements (the original Figure 4 was changed to Figure 3—figure supplement 3 and the original Figure 9 was changed to Figure 8—figure supplement 1), the figure numbers mentioned in both the reviews and the author responses do not correspond to the figures in the final article. Figures 5-8 now correspond to Figures 4-7 and Figures 10-13 now correspond to Figures 9-11.]

Thank you for submitting your article "Deciphering caveolar functions by *syndapin III* KO-mediated impairment of caveolar invagination" for consideration by *eLife*. Your article has been reviewed by three peer reviewers, one of whom, Suzanne Pfeffer, is a member of our Board of Reviewing Editors, and the evaluation has been overseen by Andrea Musacchio as the Senior Editor.

The reviewers have discussed the reviews with one another and the Reviewing Editor has drafted this decision to help you prepare a revised submission.

Summary

In their study, the authors analyzed the physiological function of syndapin III (also called PACSIN III), which they found to localize to caveolae in cardiomyocytes and muscle. For this, they generated a syndapin III knockout (KO) mouse model and investigated the morphology of caveolae in heart and muscle sections. Surprisingly, syndapin III KO myocytes revealed a severe reduction of caveolae. However, the authors could not find a severe cardiac or muscle phenotype, as described for caveolin3 knockout mouse models, which essentially lack caveolae in muscle and heart. The authors suggest that the syndapin III KO mouse can be used as a tool to dissect the function of caveolae independently from the function of caveolin3 and/or cavin1.

This is an extensive manuscript which could be very important for the caveolae field and beyond, since the lack of muscular caveolae in syndapin III ko mice suggests an unexpected function of syndapin III in the generation of caveolae. However, additional experiments will add much to the potential significance of the story, and are essential because the results presented here are so unexpected. Furthermore, there are numerous experiments for which the presentation can be improved.

Essential revisions

The reviewers were not yet completely convinced that the proposed model of syndapin-III as a curvature generator for caveolae is correct because there are still some normally shaped caveolae left in the ko animals; they thought that a BAR domain containing protein might stabilize the neck of a membrane invagination, as many other BAR domain proteins do, rather than generate the invagination. To address this issue, the reviewers suggest inclusion of some light microscopy-based experiments to quantify caveolin and cavin levels at the plasma membrane, at the plasma membrane in punctae, and at intracellular membranes in wt and ko cells. If possible, they felt that monitoring the dynamics of caveolae by transfecting cardiomyocytes with a caveolin3-GFP construct should be a rather straightforward experiment and would add a great deal to the present story (as long as it is expressed at low levels).

The reviewers have gone to great lengths to suggest specific manuscript improvements, thus I am attaching the detailed comments in full, to aid in the revision process.

1) The authors failed to mention that a model for syndapin II within caveolae was already described by Senju et al., 2015 and Hansen et al., 2011. Both studies showed that PACSIN II binds to EHD2 and is therefore recruited to caveolae, most probably to stabilize caveolar structures within the plasma membrane and not as trigger for the invagination process during caveolae generation. Hansen et al. also found a reduction of caveolae after knockdown of PACSIN II, but, in contrast to the present study, an increase of caveolin and cavin proteins in the membrane whereas in the current study, only cavin1 staining was increased in EM stainings of syndapin III KO cells. Thus, instead of invoking a decreased formation of caveolae in the absence of syndapin III, one could explain the syndapin III ko phenotype also by an increased uptake of caveolae in the absence of syndapins III. The authors should address this issue by directly looking at caveolar dynamics in primary wt and syndapins III ko cardiomyoctes or muscle cells overexpressing caveolin-3.

2) Figure 3: A double staining of syndapin III and caveolin3 was used to prove the localization of syndapin III to caveolae. However, in the KO example picture (i), it is difficult to see the cav3 staining. More micrographs, especially for the KO, and the same magnifications for wt and KO should be used to support the result. Furthermore, the authors found no significant difference between WT and KO cav3 staining at the plasma membrane (m). However, electron microscopy may not be the best method to quantitatively evaluate protein densities if different cellular structures (e.g. invaginations versus flat membranes) with potentially different accessibilities of the antibody epitopes are stained. As described in point 1, some light microscopy-based methods would be desirable to support the claims of the authors. Are Cav3 and cavin 1 still clustered?

3) Surface invaginations, caveolae, and syndapin III.

The authors use freeze-fracture and immunogold analysis to detect caveolae and the presence of syndapin III and caveolin3 at invaginations. A number of quantifications are shown, but rather than absolute quantifications, only relative quantifications between wt/ko are shown. What should in addition be shown are

- The fraction of invaginations positive for cav3 over total invaginations (are all total invaginations actual caveolae?

- The fraction of invaginations positive for sdp III over total invaginations

- The fraction of invaginations positive for both cav3 and sdp3 over total invaginations.

The cell types used in Figure 3, Figure 9 and Figure 12 might have other invaginations (clathrin-coated, non-coated), whose abundance should not be affected by Sdp III KO.

Since EM techniques allow only for fractions of cell surface areas to be examined, it would be informative to additionally determine the fraction of Sdp III-positive cav1- or cavin1-positive puncta on the surface of cultured cells using immunofluorescence of endogenous proteins (as in #2 above).

4) Caveolar and non-caveolar functions of caveolar coat components.

Knockout of Syndapin III/Pacsin3 causes loss of morphological caveolae, but not of caveolins or cavins. Based on this finding, the authors strongly emphasize that this situation uniquely allows them to disentangle the regulatory/signaling functions of caveolar coat components from the physiological role of caveolae as invaginated membrane structures. However, in this study they do not directly compare syndapin III/Pacsin3 knockout mice with caveolin3 knockout mice side-by-side and rely for their comparison on data from other studies. The authors state that "… syndapin III KO skeletal muscles showed pathological parameters upon physical exercise that are also found in CAVEOLIN3 mutation-associated diseases.", i.e. the effects of syndapin III KO are similar to a loss of caveolin3 function. We are surprised that such strong emphasis is placed on the separation of the functions of caveolar invaginations vs coat components, when no dramatic differences were found. This point could be addressed by text changes alone.

5) DRM association of dystrophin and dystroglycan (Figure 8).

The authors cite previous work that loss of caveolin3 has been associated with loss of DRM association of the dystrophin-glycoportein complex and therefore aimed at testing whether syndapin III KO caused a similar defect. In their experiments, however, even in wt animals, neither dystrophin nor β-dystroglycan floated on density gradients (Figure 7). Since there was no DRM association in lysates from WT tissues, it is impossible to determine or conclude whether syndapin III is necessary for DRM association of dystrophin or β-dystroglycan. DRM association depends on the sample/detergent ratio, the precise lysis conditions, and is therefore difficult to interpret, even more so when no DRM association is observed. In our view panels c-f do not provide significant insight and should be removed from the manuscript.

6) DRM association of cav3 and flotillin (Figure 6).

Figure 6 shows that Sdp III KO causes a change in flotation behavior of caveolin3 where the floating fraction(s) shift by two fractions towards the bottom of the gradient in Sdp III KO tissues: From fraction 3 to 5 in skeletal muscle, and from 4-6 in the heart. Flotillin1, which should not be associated with caveolar membranes, also shifts by two fractions towards the bottom of the gradient in KO tissues. The authors find support for their model that Sdp III is required specifically for organizing cav3-containing membrane domains. One could equally conclude that Sdp III is required for organizing flotillin1-containing membrane domains. Could the authors explain why they are convinced Sdp III is not involved in organizing flotillin1-containing membrane domains?

However, rather than questioning the specific role of Sdp III in caveola formation, to me this demonstrates how little can be concluded from DRM association assays. Flotillin1 might associate with caveolin-containing membranes post lysis during the centrifugation and its co-floating and shifting with cav3 may not have any biological meaning. Figure 6D-I in our view add little information to the manuscript. In the future (not required here), a more informative way for assessing the assembly state of the caveolar coat could be velocity gradients as used by the Helenius and Nichols labs (PMID: 20070607, PMID: 24013648).

7) The manuscript contains many experiments, but the organization is sometimes confusing and difficult to follow. For example, the authors often jump in between tissue and cell culture experiments for heart and muscle. It may be easier to show the immuno-stainings of syndapins III in heart and muscle directly behind the cardiomyocyte stainings (e.g. combine Figure 1, 4, 11, 12). It would also be easier to first describe the localization of syndapin III at caveolae before showing the reduced caveolar numbers in syndapin III ko cells. The previously described localization of syndapin II at caveolae should be mentioned in the Introduction.

Figure 1: The immunofluorescence experiments in C-E and I are difficult to recognize, the size of the pictures should be increased. Also the pictures in P and Q are difficult to analyze – it would be helpful if colocalisation of syndapin III and caveolin3 was investigated in more detail, e.g. using increased magnification during confocal microscopy, colocalisation analysis within ImageJ or ZEN software (see above).

Figure 2: As the authors acknowledge, the displayed EM pictures are of rather bad quality, so that caveolae cannot be precisely distinguished. Are these pictures essential?

Figure 2: The western blots for syndapin I (D) are difficult to interpret because the GAPDH signal used as loading control is very weak and no GAPDH band can be detected for the heart lysates – the authors used 50 ug/lane, so there should be a very strong signal for GAPDH. Further, the number of investigated WT and KO mice used for the western blot analysis is missing in the figure legends.

Figure 5: In G/H, the colocalization of cavin1 and caveolin3 is difficult to see. Further, the authors mentioned clustered cavin1 protein which is also not well seen in the two examples. The graphs A-F can be reduced in size and more detailed EM labeling pictures for both wt and KO can be shown in the supplement to support the results.

Figure 6: The 3D reconstruction images show the localization of syndapin III within caveolae. The authors therefore claim that syndapin III is responsible for the invagination process. However, for this model, it is quite surprising to see that only two molecules of syndapin III are detected, and only at one side/part of the caveolae. How can invagination be supported by syndapin III, if it is not found around the whole neck of caveolae? The authors should comment on this in their discussion.

Figure 8: The authors wrote in the figure legend: ' not all described cav3 loss-of-function[…]' If we understand the results described in this study, none of the previously published cav3 KO phenotypes was found in the syndapin III KO mouse?

Figure 11: Picture G, KO heart section is not well chosen, because the right ventricle is somehow cut or enrolled.

Figure 13: The muscle cell size in KO sections (C) seems to be increased. Did the author further analyze this?

Discussion:

In the Introduction, the authors suggest that caveolae and caveolin 3 function can now be better dissected, but the discussion for this is quite vague and could be extended (e.g. what do we learn?). Do the authors envisage a similar scenario for caveolin1-related caveolae in different tissue in relation to PACSIN II?

Materials and methods:

Statistical analysis should be described in more detail in the Materials and methods part, e.g. which statistical test and methods were used (t -test vs. Mann-Whitney-U test, distribution analysis of the data sets,..).

Legends for Video 1 and Video 2 refer to the wrong figure (it should be Figure 6, not Figure 5).

---

## [Author Response]

Essential revisionsThe reviewers were not yet completely convinced that the proposed model of syndapin-III as a curvature generator for caveolae is correct because there are still some normally shaped caveolae left in the ko animals; they thought that a BAR domain containing protein might stabilize the neck of a membrane invagination, as many other BAR domain proteins do, rather than generate the invagination. To address this issue, the reviewers suggest inclusion of some light microscopy-based experiments to quantify caveolin and cavin levels at the plasma membrane, at the plasma membrane in punctae, and at intracellular membranes in wt and ko cells.

We thank you for this positive assessment. We are happy to – besides the plethora of controls and comparisons already presented in the initial manuscript – provide further evidence that the fact that *syndapin III* KO leads to disruption of caveolar invagination of caveolae is not due to a reduction in cav3 levels (as it has been described for other caveolin coat-associated proteins) in our revised manuscript, which we herewith would like to submit.

We understand that beyond the classical thin sectioning and transmission electron microscopy and the biochemistry, which we used as additional techniques in parallel to our more innovative approach to label and quantitatively analyze caveolae by freeze-fracturing and TEM, the reviewers would like to see some more standard corroborative immunofluorescence analyses. We agree that this even further solidifies our study. The revised manuscript now contains several of such analyses.

Although the resolution of light microscopy is of course not sufficient to visualize single caveolae (which are only 70 nm in diameter, i.e. way below the resolution of light microscopy) and the membrane topologies within the nanodomains showing clustered cav3 signal cannot be visualized either, the lack of information on individual sites is compensated for by the power of signal averaging and intensity measurements. The revised manuscript now also by light microcopy shows that - fully consistent with our quantitative EM data in cardiomyocytes, in heart and in skeletal muscles and the biochemical evaluations – cav3 levels did not change upon *syndapin III* KO. Imaging of endogenous cav3 in sections of muscle tissue demonstrate that neither at the plasma membrane, intracellularly nor directly within puncta at the plasma membranes there were any changes in cav3 levels (newly added quantitative data panels in revised Figure 1 and Figure 12).

In addition, we were also able to answer the reviewers questions about the frequency of labeling overlap using 3D surface renderings of image stacks obtained by confocal microscopy (newly added Figure 12M, N). However, since this information is much more powerfully addressed by ultra-high resolution studies, the revised manuscript now additionally contains quantitative EM data on this (newly added Figures 3J, M and Figure 12C, D).

Furthermore, we also quantitatively analyzed the distribution of cav3 and of CAVIN-1 in detail in order to be able to describe how *syndapin III* KO affects the abundance and organization of caveolar coat components (newly added Figure 3P, Figure 5J-L and Figure 12S, T).

If possible, they felt that monitoring the dynamics of caveolae by transfecting cardiomyocytes with a caveolin3-GFP construct should be a rather straightforward experiment and would add a great deal to the present story (as long as it is expressed at low levels).

As discussed above, we agreed to corroborate and accordingly expanded our mostly EM work by additional light microscopy data in the revised manuscript. However, we felt that this is most informative in the real physiological context and therefore concentrated on examining endogenous cav3 and syndapin III in skeletal muscle tissues of WT and *syndapin III* KO mice, in which we subsequently also reveal phenotypes related to caveolinopathies in human patients.

The analyses of overexpressed GFP-cav3 in some cultured cells seemed not advisable as this would be scientifically questionable, as for example Hayer et al., 2010 (J. Cell Biol.) have revisited and extensively analyzed live imaging approaches and documented that life-imaging with GFP-caveolin leads to all sorts of artifacts, mainly to aberrant localizations of the fusion protein to non-caveolar surface pools as well as to different organelles of endosomal nature. Since the authors explicitly stated that even low expression (only 4-5 h after transfection!) is enough to cause these mislocalizations, we omitted work in this direction but rather concentrated on the endogenous protein and on the physiologically relevant situation in skeletal muscles.

We hope that you as editors and the reviewers will be content with the additional light microscopy imaging data of cav3 and syndapin III in muscles, which we added to the revised manuscript

The reviewers have gone to great lengths to suggest specific manuscript improvements, thus I am attaching the detailed comments in full, to aid in the revision process.

Thank you very much. We carefully went through all of them and improved the manuscript accordingly. Besides the above mentioned additional analyses and text changes, this also led to an increase of Figure Supplements from previously 1 to now 7 (plus 2 videos). We hope that you and the reviewers will appreciate our efforts and enjoy reading the improved manuscript.

1) The authors failed to mention that a model for syndapin II within caveolae was already described by Senju et al., 2015 and Hansen et al., 2011. Both studies showed that PACSIN II binds to EHD2 and is therefore recruited to caveolae, most probably to stabilize caveolar structures within the plasma membrane and not as trigger for the invagination process during caveolae generation.

We would like to refer the reviewer to the text of our manuscript. Although exclusively focusing on syndapin II in cell lines and not on the muscle-enriched isoform syndapin III, Hansen et al., 2011 was of course cited (introductory start of results). Careful reading of the paper published by Hansen et al., however, makes clear that these authors did not favor a caveolae stabilization model. Quote: *“Rather, pacsin 2 appears to be specifically required for these microdomains to adopt the characteristic curved caveolar membrane profile”*. Work of the Suetsugu lab on syndapin II was cited as well (Senju et al., 2011) in our quite extensive literature list (61 ref.).

We are aware of the fact that syndapin II binds to EHD2 and may thereby be recruited to caveolae; thank you for this note. Syndapin III, however, is different. As we have already shown in 2005 (Braun et al., 2005), syndapin I and II interact with EHD proteins but syndapin III does not because it lacks the required binding interface. This information is now included in the revised manuscript (including the additional references on this).

Hansen et al. also found a reduction of caveolae after knockdown of PACSIN II, but, in contrast to the present study, an increase of caveolin and cavin proteins in the membrane whereas in the current study, only cavin1 staining was increased in EM stainings of syndapin III KO cells.

Careful reading of the Hansen et al., 2011 paper leaves one with a picture more complicated than summarized by the reviewer, it seems. Hansen et al. indeed described some reduction of caveolae (defined as morphological profiles resembling caveolae in 2D) upon Pacsin2 RNAi. In contrast to that, however, the mean fluorescence intensity of the anti-cav1 and of the anti-cav2 staining per cell in TIRF images was reported as increased in the portion of RNAi cells selected for analysis due to their lower Pacsin2 epifluorescence. However, there was no statistical significance reported despite a 2fold (cav2) to 3fold (cav1) increase (Figure 6 in Hansen et al., 2011).

The CAVIN-1 data (Figure 7 in Hansen et al., 2011) is even harder to interpret, as it is presented as cav1/CAVIN-1 ratio. The curve maximum of intensity distributions per cell was shifted from about 0.33 in control to 0.42 in RNAi cells – thus by about a third (yet, this increase was described as decrease in the legend and text) – again lacking a statistical significance. It furthermore remains somewhat unclear to the reader what the cav1/CAVIN-1 ratio increase from 0.33 to 0.42 means for the absolute summed up CAVIN-1 signal per cell and for the anti-CAVIN-1 signal per cell membrane area (density) considering that cav1 levels were increased by factor 3. From this, it has to be concluded that there must have been more CAVIN-1 because with factor 3 more cav1 the maximum in the cav1/CAVIN-1 distribution curve (Figure 7 in Hansen et al., 2011) would have had to rise to about 0.9 instead of only reaching 0.42 in Pascin2 RNAi cells. Since the findings reported in Hansen et al., 2011 are difficult to report and interpret, the revised manuscript now very carefully describes and cites these particular observations of Hansen et al., 2011. We hope that the reviewer is content with the description and wording.

Apart from that, it has to be noted that Hansen et al. studied Pacsin2/syndapin II in HeLa and NIH3T3 cells and not syndapin III in muscle cells, as we did. The reviewer pointed out him/herself, that syndapin II will use molecular mechanisms completely different from those of syndapin III, as syndapin II can interact with the caveolar component EHD2, whereas syndapin III is unable to associate with EHD proteins (Braun et al., 2005).

Thus, instead of invoking a decreased formation of caveolae in the absence of syndapin III, one could explain the syndapin III ko phenotype also by an increased uptake of caveolae in the absence of syndapins III. The authors should address this issue by directly looking at caveolar dynamics in primary wt and syndapins III ko cardiomyoctes or muscle cells overexpressing caveolin-3.

We have carefully thought about the idea of the reviewer. It remained somewhat unclear to us how to relate the idea of a putatively increased uptake to the fact that we observed constant levels or even slightly increased levels of cav3 and CAVIN-1 at the plasma membrane (which are in line with increased or similar levels of cav1 and CAVIN-1 in HeLa cells described by Hansen et al. 2011).

Furthermore, one has to take into account two aspects: First, TIRF formally does not show internalization (which in other cell biological uptake processes e.g. can be proven by pH-sensitive tools or assays tracking chemical labeling access from the outside) but just addresses presence or disappearance from a TIRF zone, which is about 100-200 nm in depth. It thus usually remains open whether the fluorescence signal dissolves in the membrane (lateral dispersion of coat components), complexes disassemble in 3D (i.e. including dispersion into the cytosol), or whether a given signal leaves the TIRF zone by trafficking of some membrane compartment. Also, it remains unclear what the start situation is. Since the width of the PM is only 6-10 nm, not 200 nm, cortically localized, vesicle-docked components can reside in a TIRF zone without ever having been part of the PM.

Second, analyses of overexpressed GFP-cav3 seemed not advisable for two reasons: i) Hayer et al. 2010 (J. Cell Biol.) have revisited and extensively analyzed live imaging approaches and documented that live-imaging with GFP-caveolin leads to all sorts of artifacts, mainly to aberrant localizations of the fusion protein to non-caveolar surface pools as well as to different organelles of endosomal nature. The authors explicitly stated that even low expression (only 4-5 h after transfection!) is enough to cause these mislocalizations. ii) The low transfection rates achievable with primary cardiomyocytes or even with cultured muscle cell, as suggested by the reviewer, would lead to low n-numbers of cells accessible for analysis.

We thus rather concentrated on the endogenous protein and on the physiologically relevant situation in skeletal muscles to further address the idea of the reviewer that caveolae internalization is changed upon *syndapin III* KO and analyzed the intensity of intracellular cav3. The intracellular cav3 levels in muscle cells turned out to be unchanged, as addressed by quantitative immunofluorescence analyses of skeletal muscle sections (newly added Figure 12I of the revised manuscript).

These data are in line with the biochemical and EM data presented in our manuscript. The findings that the Triton-insoluble pool of cav3 was not changed (Figure 11R, 12K) and that the plasma membrane-associated pool was not changed in *syndapin III* KO animals (Figure 3O, 11Q and 12H,L) can only be related to the also unchanged general expression levels of cav3 (Figure 5), if also the intracellular pool of cav3 was unchanged.

2) Figure 3: A double staining of syndapin III and caveolin3 was used to prove the localization of sydapin III to caveolae. However, in the KO example picture (i), it is difficult to see the cav3 staining. More micrographs, especially for the KO, and the same magnifications for wt and KO should be used to support the result.

As documented by the bars (in both h and i panels 100 nm), the magnifications of WT and KO are identical.

We acknowledge that the anti-cav3 labels (10 nm gold) in Figure 3H and I – although highlighted by arrowheads – may have been hard to see. We have therefore now highlighted them with blue color (and the anti-syndapin III labels with red color) (revised Figure 3H,I).

We hope that the reviewer will agree with us that this makes it much easier to see the localization of cav3 and cav3 clusters as well as the cav3/syndapin III colocalization at caveolar structures, at shallow membrane indentations and at flat membrane areas. The unlabeled raw data sets are presented in the additionally added Figure 3—figure supplement 2 of the revised manuscript.

As far as some more data on colocalization of syndapin III with cav3 is concerned, we would also like to refer the reviewer to the quantitative analyses of confocal images of transversal muscle sections in the newly added Figure 1Q of the revised manuscript and to the analyses of longitudinal sections shown in the newly added Figure 12E-J. Furthermore we specified the frequency of colabeling at cav3 puncta in such longitudinal sections at the light microscopical level (newly added Figure 12M,N). Furthermore, we used quantitative electron microscopy to work this out in more detail (newly added Figure 3J (cardiomyocytes) and Figure 12C (skeletal muscle) of the revised manuscript).

Since the reviewer also requested to show even larger membrane areas to document the reduction of cav3-positive caveolar invaginations upon *syndapin III* KO (already the previous manuscript presented an enlarged membrane area when compared to WT to visualize this finding of our quantitative analyses), we furthermore added another KO picture to Figure 3I so that the revised Figure 3 now contains the panels I and I’. Besides this addition to the main manuscript, we also added the WT image of the large membrane area, from which the small insets (Figure 3H’ and H’’) have been taken, to the Supplements (newly added Figure 3—figure supplement 1).

Furthermore, the authors found no significant difference between WT and KO cav3 staining at the plasma membrane (m). However, electron microscopy may not be the best method to quantitatively evaluate protein densities if different cellular structures (e.g. invaginations versus flat membranes) with potentially different accessibilities of the antibody epitopes are stained. As described in point 1, some light microscopy-based methods would be desirable to support the claims of the authors.

As the reviewer considered it important to corroborate the findings we obtained by sophisticated EM methods not only by standard EM techniques and biochemical examinations but also by further light microscopical analyses, the revised manuscript now further strengthens our results with several light microscopical evaluations: The cav3 levels at the plasma membrane of muscle cells remained identical upon *syndapin III* KO and the intracellular cav3 levels also remained identical upon *syndapin III* KO (consistent with EM and quantitative biochemical examinations showing similar general expression and Triton-resistant levels of cav3) (newly added Figure 12H,L of the revised manuscript).

Quantitative, confocal analyses of longitudinal muscle sections furthermore revealed that also the cav3 content in “puncta” at the plasma membrane was similar in WT and syndapin KO muscle fibers (newly added Figure 12J of the revised manuscript).

In general, different accessibilities of antibodies at different places could of course always be a problem. This criticism of the reviewer, however, applies to all antibody-based methods in life sciences.

Furthermore, immunofluorescence labeling and imaging due to its low sensitivity for single molecules and due to its low resolution (especially in Z) is susceptible to thresholding and 3D artefacts.

More classical imaging methods, such as confocal microscopy, TIRF or even TEM of sections fail to firmly prove that an antibody label, i.e. the protein of interest, is not just by chance in the vicinity of the membrane (confocal, about 100-300 nm; TIRF area, 100-200 nm; TEM, resolution easily reaches 1-2 nm but antibody extensions are about 20 nm and no selection is obtained at the level of sample preparation). As explained in the manuscript, these disadvantages are circumvented by freeze-fracturing (only membrane-inserted material remains) and immunogold labeling (excellent resolution in Y,X and by shadowing, table tilting and/or tomography also good resolution in Z). Thus, quantitative analysis of immunogold labeled, freeze-fractured replica is a very accurate method of analysis.

Are Cav3 and cavin 1 still clustered?

As mentioned in the revised manuscript, Figure 3I, Figure 4F and Figure 5H suggest that this still is the case for both cav3 and CAVIN-1 at the plasma membrane in cardiomyocytes. More importantly, also in the tissue context (Figure 11N (heart) and Figure 12B (skeletal muscle)), cav3 clusters prevail upon *syndapin III* KO.

As this aspect may be of some mechanistic importance, we conducted a detailed cav3 cluster analysis directly in tissue samples of skeletal muscles. Interestingly, *syndapin III* KO led to a small (-7%) but significant drop of the density of cav3 within clusters accompanied with a reduction of cav3 clusters at the plasma membrane. Thus, syndapin III either directly or indirectly seems to have some importance for cav3 clustering, too (newly added Figure 12S,T).

Additional CAVIN-1 analyses at freeze-fractured muscle membranes unfortunately were not possible for technical reasons and/or CAVIN-1 in muscle cells shows too low expression. In fact, confocal immunofluorescence analyses of skeletal muscles suggested that CAVIN-1 expression is low at plasma membranes of muscle cells suggesting that a large part of the anti-CAVIN-1 immunosignal that one can e.g. see in western blotting of muscle samples predominantly may actually stem from satellite and endothelial cells (our unpublished results). We were, however, able to address the clustering of CAVIN-1 in dissociated *syndapin III* KO cardiomyocytes. Importantly, CAVIN-1 clusters clearly remained present in *syndapin III* KO and the density of CAVIN-1 immunolabels in the clusters increased accordingly to the overall increase of CAVIN-1 at the plasma membrane (newly added Figure 5J). Thus, even clustered cav3 and CAVIN-1 at the plasma membrane is not sufficient for bringing about caveolae, if syndapin III is absent.

In fact the CAVIN-1 cluster densities even increased (newly added Figure 5K). This may be caused by the increased levels of CAVIN-1 at the plasma membrane already described in the previous version of the manuscript (Figure 5I). An increased percentage of the more abundant CAVIN-1 clusters in *syndapin III* KO cardiomyocytes, however, were cav3-negative (newly added Figure 5L).

3) Surface invaginations, caveolae, and syndapin III.The authors use freeze-fracture and immunogold analysis to detect caveolae and the presence of syndapin III and caveolin3 at invaginations. A number of quantifications are shown, but rather than absolute quantifications, only relative quantifications between wt/ko are shown. What should in addition be shown are- The fraction of invaginations positive for cav3 over total invaginations (are all total invaginations actual caveolae?

The revised manuscript now shows that 58% of all invaginations in cardiomyocytes are anti-cav3-labelled. It also shows that only 9% are non-caveolar structures (newly added Figure 3J,M of revised manuscript).

In skeletal muscles, non-caveolar structures only account for 1% of all invaginations. 96% of all invaginations are anti-cav3 labelled (newly added Figure 12C,D of revised manuscript).

- The fraction of invaginations positive for sdp III over total invaginations

In cardiomyocytes, 91% of all invaginations are deep caveolae and shallow indentations of caveolar appearance marked by cav3. 58% of all invaginations are syndapin III-positive and have caveolae-like (deep and shallow) profiles (the majority of them also proven as caveolae by associated cav3 signals) (newly added Figure 3J).

An additional 1% are anti-syndapin III-positive non-caveolar structures (newly added Figure 3J and M).

In skeletal muscles, anti-syndapin III labeling is less effective but still 20% of all invaginations (99% caveolar; 96% further proven by cav3 association) are syndapin III-positive deep and shallow caveolar profiles.

An additional 0.5% represent syndapin III-positive invaginations of non-caveolar nature (newly added Figure 12C and D)

- The fraction of invaginations positive for both cav3 and sdp3 over total invaginations.The cell types used in Figure 3, Figure 9 and Figure 12 might have other invaginations (clathrin-coated, non-coated), whose abundance should not be affected by Sdp III KO.

Double labelings are of course usually not so easy at the high resolution of EM but 35% of all caveolae are double-positive for sdp3&cav3 in cardiomyocytes (newly added Figure 3J). In muscles, 19% of all invaginations are double-positive for syndapin III and cav3, i.e. almost all syndapin III-positive structures also are positive for cav3 (newly added Figure 12C).

In both cases, all of these double-labelled invaginations are of caveolar morphology (newly added Figure 3J and 12C).

Additional analyses at the light microscopical level confirmed this. 32% of all cav3-enriched fields/puncta at the plasma membrane of longitudinal muscle sections were positive for syndapin III (newly added Figure 12M,N).

Hansen et al., 2011 reported that in cell lines “Co-staining of pacsin 2 and caveolin 1 revealed that many, but not all, caveolin 1 puncta also contain pacsin 2 (Figure 2A). Approximately 35% of caveolin 1 puncta were positive for pacsin2”. Thus, our quantitative data for muscle tissues and cav3/syndapin III seem to be consistent with the estimation reported for the syndapin II isoform and cav1 in non-muscle cells at the light microscopical level.

The (very low) abundance of the non-caveolar invaginations we detected was not significantly affected by *syndapin III* KO. In both cardiomyocytes and skeletal muscle tissue, there was no statistically significant difference between WT and KO (newly added Figures 3M and 12D).

Since EM techniques allow only for fractions of cell surface areas to be examined, it would be informative to additionally determine the fraction of Sdp III-positive cav1- or cavin1-positive puncta on the surface of cultured cells using immunofluorescence of endogenous proteins (as in #2 above).

The reviewer probably means cav3 and not cav1, as we have demonstrated that cav1 and syndapin III are not coexpressed. Figure 4 shows that cav1 is only present in non-muscle cells and cav3 and syndapin III are expressed in muscle cells.

In case of cav3, as already described above, we conducted such immunofluorescence analyses during our revision work and observed that at the level of light microscopical resolution 32% of all anti-cav3 puncta are immunopositive for endogenous syndapin III in confocal microscopy analyses of longitudinal muscle sections (please see newly added Figure 12M,N). As also already mentioned above, this is somewhat reminiscent of the estimation reported by Hansen et al., 2011 for syndapin II/Pacsin2 and cav1.

4) Caveolar and non-caveolar functions of caveolar coat components.Knockout of Syndapin III/Pacsin3 causes loss of morphological caveolae, but not of caveolins or cavins. Based on this finding, the authors strongly emphasize that this situation uniquely allows them to disentangle the regulatory/signaling functions of caveolar coat components from the physiological role of caveolae as invaginated membrane structures. However, in this study they do not directly compare syndapin III/Pacsin3 knockout mice with caveolin3 knockout mice side-by-side and rely for their comparison on data from other studies. The authors state that "… syndapin III KO skeletal muscles showed pathological parameters upon physical exercise that are also found in CAVEOLIN3 mutation-associated diseases.", i.e. the effects of syndapin III KO are similar to a loss of caveolin3 function. We are surprised that such strong emphasis is placed on the separation of the functions of caveolar invaginations vs coat components, when no dramatic differences were found. This point could be addressed by text changes alone.

We hope that the changes we made to the revised manuscript will help to prevent such misunderstandings. The skeletal muscle defects mirror cav3 defects, i.e. indeed reflect the loss of caveolar functions.

In contrast, e.g. in the heart and in the case of ERK1/2 signaling, the results of *syndapin III* KO – despite the strong reduction of caveolar structures – do NOT mirror the effects of caveolin deficiency. This strongly suggest that those impairments observed upon caveolin deficiency rather reflect a lack of caveolin at the membrane (or are indirect consequences of this) rather than reflecting a loss of caveolar structures and their functions. Thus, here we can for the first time dissect biological functions of caveolar invagination from those merely reflecting caveolin presence. For understanding the pathophysiology of caveolinopathies this is of utmost importance. Also, it may focus therapeutic efforts at parameters one may actually have a chance of influencing.

5) DRM association of dystrophin and dystroglycan (Figure 8).The authors cite previous work that loss of caveolin3 has been associated with loss of DRM association of the dystrophin-glycoportein complex and therefore aimed at testing whether syndapin III KO caused a similar defect. In their experiments, however, even in wt animals, neither dystrophin nor β-dystroglycan floated on density gradients (Figure 8 C,E). Since there was no DRM association in lysates from WT tissues, it is impossible to determine or conclude whether syndapin III is necessary for DRM association of dystrophin or β-dystroglycan. DRM association depends on the sample/detergent ratio, the precise lysis conditions, and is therefore difficult to interpret, even more so when no DRM association is observed. In our view panels c-f do not provide significant insight and should be removed from the manuscript.

The reviewer is right. Indeed we found it quite surprising that the DRM association of dystrophin and β-dystroglycan reported in the literature obviously is not robust and therefore a hard to interpret caveolin loss-of-function phenotype. Although we think that the reviewer will agree with us that the results we obtained were correctly described in the previous version of the manuscript, we omitted these findings from the revised manuscript, as the reviewer requested (please see revised Figure 8).

6) DRM association of cav3 and flotillin (Figure 7).Figure 7 shows that Sdp III KO causes a change in flotation behavior of caveolin3 where the floating fraction(s) shift by two fractions towards the bottom of the gradient in Sdp III KO tissues: From fraction 3 to 5 in skeletal muscle, and from 4-6 in the heart. Flotillin1, which should not be associated with caveolar membranes, also shifts by two fractions towards the bottom of the gradient in KO tissues. The authors find support for their model that Sdp III is required specifically for organizing cav3-containing membrane domains. One could equally conclude that Sdp III is required for organizing flotillin1-containing membrane domains. Could the authors explain why they are convinced Sdp III is not involved in organizing flotillin1-containing membrane domains?However, rather than questioning the specific role of Sdp III in caveola formation, to me this demonstrates how little can be concluded from DRM association assays. Flotillin1 might associate with caveolin-containing membranes post lysis during the centrifugation and its co-floating and shifting with cav3 may not have any biological meaning. Figure 7D-I in our view add little information to the manuscript. In the future (not required here), a more informative way for assessing the assembly state of the caveolar coat could be velocity gradients as used by the Helenius and Nichols labs (PMID: 20070607, PMID: 24013648).

We thank the reviewer for his/her advice on future biochemical examinations using velocity gradients instead of classical DRM association assays.

We agree with the reviewer that DRM association assays indeed often are difficult to interpret and that flotillin associations even may be post lysis. Although this would still suggest that some alterations of the protein/lipid composition and/or the organization of cav3-containing membrane domains occur upon *syndapin III* KO (and only this point was made by the flotillin observation in the previous manuscript), we followed the advice of the reviewer and let out the flotillin data to not confuse readers (revised Figure 7D-G).

Whether the flotillin effects are post-lysis artifacts, are indirectly somehow dependent on cav/lipid/syndapin III nanodomains or are independently caused by *syndapin III* KO, can currently not be answered and would be way beyond the scope of the current manuscript focusing on caveolar invagination and caveolar function in vivo.

7) The manuscript contains many experiments, but the organization is sometimes confusing and difficult to follow. For example, the authors often jump in between tissue and cell culture experiments for heart and muscle. It may be easier to show the immuno-stainings of syndapins III in heart and muscle directly behind the cardiomyocyte stainings (e.g. combine Figure 1, 4, 11, 12).

The reviewer is right that our manuscript contains a lot of data and it is acknowledged that some of the figures, such as Figure 4 and 9 (maybe even Figure 8 and 11 as well) in other high-profile journals only accepting fewer main figures could probably have been supplementary figures rather than integrated into the main manuscript as it is the policy of *eLife*.

Our report starts with the discovery of a striking cell biological phenotype of a new KO mouse and then in detail and step by step unveils the mechanistic principles behind these impairment to provide a full picture of the cell biological and molecular details. Finally, after the cell biology is understood, the manuscript then reports on examinations addressing the physiological relevance at the animal level. In your eyes, this design of our study and this structure of our manuscript is very much appropriate for a high-ranking journal, such as *eLife*, which publishes comprehensive studies on important cell biological processes.

To organize the data a little bit more according to the cellular system analyzed and less according to scientific question, as the reviewer suggested, we have moved the biochemical examinations of cav3 in heart and skeletal muscles, which previously was part of the cardiomyocyte-Figure 3 into the figures that deal with heart tissue analyses (Figure 11) and with examinations of cav3 and caveolar functions in skeletal muscle tissues (Figure 12), respectively (inserted panels Figure 11R and 12K (from Figure 3N,O).

It would also be easier to first describe the localization of syndapin III at caveolae before showing the reduced caveolar numbers in syndapin III ko cells.

The localization of syndapin III at caveolae is shown before the reduced numbers of caveolae in syndapin III cells are examined and described in detail. Please see Figure 3 and Figure 3—figure supplements and the respective Results text of the revised manuscript.

The previously described localization of syndapin II at caveolae should be mentioned in the Introduction.

We tried to move cited work on other syndapin isoforms – especially on syndapin II and cav1 – forward from Results and Discussion into the Introduction. However, the introduction focuses on CAV3 mutations and caveolinopathies and not on syndapins. Therefore, we did not find a reasonable solution for including information on the other syndapin isoforms and their similar and different functions and their different protein properties without disrupting the flow of the Introduction or expanding its length significantly. We hope the reviewer will be content with the changes we made to the revised manuscript to discuss work on syndapin II.

Figure 1: The immunofluorescence experiments in c-e and i are difficult to recognize, the size of the pictures should be increased.

All of these panels have been slightly increased in size in the revised manuscript (see revised Figure 1C-E, I). In addition, the localization of syndapin III in membrane folds and mCherry-F-marked plasma membrane domains can now be seen better in a magnified inset placed into the revised Figure 1E.

The increased Figure 1C-E and Figure 1I panels can in addition be found at significantly higher magnification in the Supplements (newly added Figure 1—figure supplement 1 and Figure 1—figure supplement 2).

Also the pictures in p and q are difficult to analyze – it would be helpful if colocalisation of syndapin III and caveolin3 was investigated in more detail, e.g. using increased magnification during confocal microscopy, colocalisation analysis within ImageJ or ZEN software (see above).

We have done this in two ways. First, we did not just run colocalization analyses on the images 1p and 1q shown, as the reviewer suggested, but conducted systematic analyses of transversal muscle sections and in total analyzed 230 PM ROIs and 160 intracellular ROIs. For the plasma membrane, the Pearson coefficient was positive and very high (0.55). Intracellularly, the correlation was just random, i.e. around zero (0.08). These quantitative analyses of the localization of cav3 and syndapin III in confocal image stacks of transversally cut skeletal muscles are now included in the revised manuscript (newly added Figure 1Q).

Second, we have analyzed the frequency of syndapin III in cav3-positive membrane domains in more detail using EM (newly added Figure 3J (cardiomyocytes) and newly added Figure 12C (skeletal muscle)) as well as by additionally employing confocal immunofluorescence microscopy of longitudinal sections of skeletal muscles to further increase the resolution of such light microscopical colocalization studies (newly added Figure 12E-G,J). Also these different colocalization studies were evaluated quantitatively.

The confocal microscopy analyses clearly showed the presence of syndapin III at a subpopulation of caveolae (Pearson coefficient for 60 puncta ROIs in longitudinal sections still positive, 0.3; newly added Figure 12E-G). Depending on the biological systems analyzed, the frequencies of syndapin III labeling also were very good. 32% of the endogenous anti-cav3 puncta were also positive for syndapin III at the lower light microscopical resolution (newly added Figure 12m,n). At the EM, level, syndapin III immunolabeling was observed at 20% of all invaginations in cryo-preserved and freeze-fractured skeletal muscles (all of caveolae-like profile; 19% also proven to represent caveolar structures by cav3 association; newly added Figure 12C) and at 59% of all invaginations in cardiomyocytes (58% caveolae-like syndapin III-positive, only 1% non-caveolar and syndapin III-positive; newly added Figure 3J).

Figure 2: As the authors acknowledge, the displayed EM pictures are of rather bad quality, so that caveolae cannot be precisely distinguished. Are these pictures essential?

We hope that also the reviewer will acknowledge that it is not the images presented that are of bad quality but that this type of analysis just has several technique-immanent limitations. This is also what we wrote in the manuscript. The pictures therefore just highlight these technique-immanent limitations.

Furthermore, TEM of sections is a standard method used in the field and we therefore had to compare our phenotypical analysis with this standard technique additionally to presenting the much more informative, quantitative analysis of large areas of membrane by immunolabeling of freeze-fractured samples – i.e. use a more sophisticated method only few labs in the world have established.

Figure 2: The western blots for syndapin I (d) are difficult to interpret because the GAPDH signal used as loading control is very weak and no GAPDH band can be detected for the heart lysates – the authors used 50 ug/lane, so there should be a very strong signal for GAPDH. Further, the number of investigated WT and KO mice used for the western blot analysis is missing in the figure legends.

GAPDH in heart indeed was very weak. We have redone this and have replaced Figure 2 by a data set, which shows GAPDH more clearly in all tissue tested (revised Figure 2D).

The missing n-number information has been added to the legends of Figure 2, Figure 5 (both n=12 each genotype) and 11 (n=6 each genotype) in the revised manuscript.

Figure 5: In G/H, the colocalization of cavin1 and caveolin3 is difficult to see. Further, the authors mentioned clustered cavin1 protein which is also not well seen in the two examples.

We acknowledge that in particular the anti-cav3 labels (10 nm gold) were hard to see in Figure 5 and H. We have therefore now highlighted them with blue color (and the anti-CAVIN-1 labels with green color) (revised Figure 5). The unlabeled raw data is presented in the additionally added Figure 5—figure supplement 1.

Some examples of CAVIN-1 clusters are marked by arrowheads.

The graphs a-f can be reduced in size and more detailed EM labeling pictures for both wt and KO can be shown in the supplement to support the results.

The graphs in A,B,D and F were reduced in sized and both the 10 nm and the 15 nm immunogold in the EM pictures were labelled in the revised Figure 5. Further images for both WT and KO have been added (see newly added Figure 5—figure supplement 1).

Figure 6: The 3D reconstruction images show the localization of syndapin III within caveolae. The authors therefore claim that syndapin III is responsible for the invagination process. However, for this model, it is quite surprising to see that only two molecules of syndapin III are detected, and only at one side/part of the caveolae. How can invagination be supported by syndapin III, if it is not found around the whole neck of caveolae? The authors should comment on this in their discussion.

Obviously much less syndapin than caveolin coat proteins are needed for invagination. This is expected because syndapin III is not a general coat component, as seen in the tomographs.

It is conceivable that induction of membrane curvature at a nanodomain (generated by syndapin III self-association) is enough to trigger caveolin coat assembly and/or bending. Once such an invagination is triggered, the coat component cav3 (presumably with the help of further factors) may then propagate this further. As requested by the reviewer, this is now discussed in more detail in the revised manuscript.

Figure 8: The authors wrote in the figure legend: ' not all described cav3 loss-of-function[…]' If we understand the results described in this study, none of the previously published cav3 KO phenotypes was found in the syndapin III KO mouse?

Please see our answer above, this may be a misunderstanding. As shown in Figure 13, *syndapin III* KO does cause phenotypes reminiscent of muscle dystrophies seen in caveolinopathies. Thus, there are certain caveolin loss-of-function phenotypes that are clinically important and mirrored very well by *syndapin III* KO.

In contrast, some other phenotypes previously also attributed to the lack of caveolae were not seen upon a loss of caveolar invaginations in *syndapin III* KO, i.e. may rather reflect a loss of caveolin membrane hubs as such but not the functions of caveolae as such. These data are summarized in Figure 8 as far as molecular/cell biological findings are concerned (in both heart and muscle). Besides the missing changes in ERK1/2 signaling, detailed analyses did not show the reported changes in the membrane association and the levels of dystrophin/β-dystroglycan (Figure 8). Figure 11 follows this up at the physiological level and finds no phenocopy in the heart (Figure 11).

Figure 11: Picture g, KO heart section is not well chosen, because the right ventricle is somehow cut or enrolled.

Indeed, the heart section that was presented in Figure 11 in the previous manuscript for left ventricle analyses was not optimal for the other, the right ventricle. We replaced Figure 11 by a new pair of images (see revised Figure 11).

Figure 13: The muscle cell size in KO sections (c) seems to be increased. Did the author further analyze this?

The caliber spectrum was analyzed in detail. We would like to refer the reviewer to Figure 13I-L.The caliber spectrum is changed in untrained *syndapin III* KO mice. These defects can be compensated for by training but this comes at the expense of detached nuclei and events of cellular damage in the tissue (Figure 13H and 13C-F).

Discussion:In the Introduction, the authors suggest that caveolae and caveolin 3 function can now be better dissected, but the discussion for this is quite vague and could be extended (e.g. what do we learn?). Do the authors envisage a similar scenario for caveolin1-related caveolae in different tissue in relation to PACSIN II?

This is now covered in more detail in the discussion of the revised manuscript.

Syndapin II is quite different from syndapin III, as for example syndapin II interacts with EHD proteins and syndapin III does not (Braun et al., 2005). EHD2 seems to play some important role for caveolae, too. Thus, it is an open question whether syndapin II has functions in cav1-positive cell systems that are similar to and as critical as those of syndapin III in muscle cells.

Apart from that, it also is an open question whether syndapin II loss-of-function phenotypes, that manifest at the cellular level (Hansen et al., 2011; Koch et al., 2012), have consequences at the physiological level that are similar and/or as clear as those observed for *syndapin III* KO in our study. To a large part, this may depend on whether membrane shaping proteins that are functionally/structurally somewhat related to syndapin II will be able to take over (some of) syndapin II’s functions in cav1 expressing cells.

As we demonstrated in our study, for syndapin III and cav3 such compensatory mechanisms seem not to exist.

Materials and methods:Statistical analysis should be described in more detail in the Materials and methods part, e.g. which statistical test and methods were used (t -test vs. Mann-Whitney-U test, distribution analysis of the data sets,..).

The statistical tests are specified for all individual figure panels in the figure legends. We also added the information that besides testing for statistical significance also testing for normal distribution of data (=> t-test) or not (=> Mann-Whitney-U) has been done with Prism6.

Legends for Video 1 and Video 2 refer to the wrong figure (it should be Figure 6, not Figure 5).

We apologize for this embarrassing error in the legends of the supplementary movies and sincerely thank the reviewer for his/her thorough review and for pointing this out. The mistake has been corrected in the revised manuscript.